# Efflux pump activity potentiates the evolution of antibiotic resistance across *S. aureus* isolates

Andrei Papkou ⬤ [1,2,4✉], Jessica Hedge ⬤ [1,4], Natalia Kapel[1], Bernadette Young ⬤ [3] & R. Craig MacLean ⬤ [1✉]

The rise of antibiotic resistance in many bacterial pathogens has been driven by the spread of a few successful strains, suggesting that some bacteria are genetically pre-disposed to evolving resistance. Here, we test this hypothesis by challenging a diverse set of 222 isolates of *Staphylococcus aureus* with the antibiotic ciprofloxacin in a large-scale evolution experiment. We find that a single efflux pump, *norA*, causes widespread variation in evolvability across isolates. Elevated *norA* expression potentiates evolution by increasing the fitness benefit provided by DNA topoisomerase mutations under ciprofloxacin treatment. Amplification of *norA* provides a further mechanism of rapid evolution in isolates from the CC398 lineage. Crucially, chemical inhibition of NorA effectively prevents the evolution of resistance in all isolates. Our study shows that pre-existing genetic diversity plays a key role in shaping resistance evolution, and it may be possible to predict which strains are likely to evolve resistance and to optimize inhibitor use to prevent this outcome.

[1] Department of Zoology, University of Oxford, 11a Mansfield Road, Oxford OX1 3PS, UK. [2] Department of Evolutionary Biology and Environmental Studies, University of Zurich, Winterthurerstrasse 190, Zurich CH-8057, Switzerland. [3] Nuffield Department of Clinical Medicine, John Radcliffe Hospital, University of Oxford, Oxford OX3 9DU, UK. [4] These authors contributed equally: Andrei Papkou, Jessica Hedge. ✉email: andrei.papkou@uzh.ch; craig.maclean@zoo.ox.ac.uk

Infections caused by antibiotic resistant bacteria are currently estimated to cause ~700,000 deaths per year, and this mortality rate is predicted to increase to 10 million per year by 2050[1]. Given this threat, resistance has been identified as one of the most important challenges to human health by a wide variety of national and international bodies, including the WHO, the G8 and the IMF. To solve this crisis, we require new antimicrobials to treat infections caused by resistant pathogens and new approaches to predict and prevent the spread of resistance in pathogen populations[2–4].

The increase in antibiotic resistance in many pathogens has been driven by the spread of a relatively small number of very successful antibiotic resistant lineages[5–10]. One explanation for this pattern is that these successful lineages are simply those that, by chance, successfully acquire rare antibiotic resistance genes by mutation or horizontal gene transfer[11,12]. Alternatively, it is possible that some strains of bacteria are more likely to evolve resistance than others, for example because they have an elevated mutation rate[13], or because they carry 'potentiator' genes that open up new genetic paths to evolving resistance[14,15]. If so, it may be possible to identify strains of bacteria that are at high risk of evolving resistance to antibiotics, and to change antibiotic usage to prevent this outcome and associated treatment failure[16].

In this paper, we investigate genomic drivers of evolvability using comparative experimental evolution. In this approach, many bacterial 'parental' strains are challenged with a common selective pressure using controlled and highly replicated in vitro evolution experiments. Following the trajectory of replicate selection lines after antibiotic exposure makes it possible to estimate the evolvability of each strain, and downstream phenotypic and genomic analyses are used to characterize evolved populations[14,17]. This approach makes it possible to address three separate questions: (i) Does the ability to evolve antibiotic resistance differ between bacterial strains? (ii) What are the underlying mechanisms of resistance evolution? (iii) How does genome content determine the rate and mechanisms of resistance evolution?

We applied this approach at a very large scale by challenging a diverse collection of 222 parental strains of *Staphylococcus aureus* with ciprofloxacin using a highly replicated selection experiment. Parental strains were sampled from a large collection of nasal carriage and bacteraemia isolates of *S. aureus* collected at hospitals in Oxford and Brighton, UK[18], and were chosen to capture a diverse set of clonal-complexes of human-associated *S. aureus*. We chose to focus on ciprofloxacin because the evolution of ciprofloxacin resistance has played a key role in the success of MRSA lineages[19], and has been associated with poor clinical outcomes for patients infected with CC22 *S. aureus*[20].

## Results

**Evolution of ciprofloxacin resistance across *S. aureus*.** We serially passaged 12 replicate populations of 222 parental strains of *S. aureus* into fresh culture medium supplemented with the clinical breakpoint concentration of ciprofloxacin (1 mg l$^{-1}$) for 5 days and recorded bacterial density over time (Fig. 1a, b and Supplementary Fig. 1). All of the parental strains were ciprofloxacin sensitive according to EUCAST clinical breakpoint (MIC < 1 mg l$^{-1}$) and we initiated our populations with an inoculum small enough (~$3 \times 10^5$ cells) to virtually guarantee that no pre-existing resistance mutations were present at the outset of the experiment (cip$^R$ mutation rate $5.1 \times 10^{-9}$ per cell division, 95% CI = [$3.7 \times 10^{-9}$, $6.7 \times 10^{-9}$]). Conventional antibiotic treatment strategies rely on using large doses of antibiotics to clear pathogen populations, and this simple experimental design seeks to capture this by creating a scenario in which populations must either evolve resistance or face extinction.

Bacterial population density declined at the start of the experiment due to the bactericidal effects of ciprofloxacin and the population bottlenecking imposed by serially passaging cultures. However, population density eventually recovered in a subset (1075 of 2664; 40%) of cultures, suggesting that these populations had expanded due to the de novo evolution of ciprofloxacin resistance during our experiment. To test this idea, we measured the ciprofloxacin MIC for 83 populations that recovered, each derived from a different parental strain (Supplementary Fig. 2). All evolved populations had an MIC > 1 mg l$^{-1}$, demonstrating that population recovery was driven by the evolution of resistance. Therefore, we measured the evolvability of ciprofloxacin resistance in the parental strains as the fraction of populations that survived until the end of the selection experiment. Our selection experiment revealed striking variation in evolvability across *S. aureus*; for example, resistance always evolved in 24 strains, whereas resistance never evolved in 39 strains (Supplementary Fig. 3a).

All else being equal, strains with a higher initial resistance to ciprofloxacin should have a greater opportunity to evolve resistance to a fixed dose of ciprofloxacin. For example, populations of strains with higher initial resistance should decline more slowly under ciprofloxacin treatment, allowing more opportunity for the emergence of novel ciprofloxacin resistance mutations[21]. To test this hypothesis, we measured the growth of the parental strains across a fine gradient of ciprofloxacin concentrations, and used growth data to calculate a ciprofloxacin MIC and IC$_{50}$ for each strain. Although the parental strains were all ciprofloxacin sensitive with respect to the EUCAST clinical breakpoint (i.e. MIC ≤ 1 mg l$^{-1}$), we found subtle and quantitative variation in initial resistance between the parental strains— for the rest of the manuscript we refer to this variation as the intrinsic resistance of the parental strains (Supplementary Fig. 4).

Evolvability increased with intrinsic ciprofloxacin resistance, and intrinsic resistance accounts for 27% of the variation in evolvability between parental strains (Pearson's correlation $r = 0.55$, $N = 222$, $P = 2e{-}18$; Supplementary Table 1). One interesting insight from this analysis is that small differences in intrinsic resistance between strains were associated with large differences in evolvability. For example, using a logistic regression we estimated that a strain with IC$_{50} = 0.12$ mg l$^{-1}$ (median across all strains) would have a 0.36 probability of evolving resistance. Decreasing or increasing the IC$_{50}$ by only 0.1 mg l$^{-1}$ (equivalent to 1/10 of the selection dose) would change the predicted evolvability to 0.10 or 0.72, correspondingly (Supplementary Fig. 3b, Supplementary Table 1, 2). In contrast, there was no correlation between the resistance (i.e. IC$_{50}$) of evolved populations and their parental strains, suggesting that the mechanisms of high level resistance that evolved during the selection experiment were independent of the intrinsic resistance of the parental strains (Pearson's correlation $r = -0.03$, $N = 83$, $P = 0.7989$).

It is also possible that some strains of *S. aureus* are genetically pre-disposed to evolving resistance, for example because they have a high mutation rate[13,22], or alternative mechanisms for evolving resistance[14]. To test if strains belonging to different lineages of *S. aureus* evolved resistance at different rates, we grouped all parental strains into 14 non-overlapping clusters based on genetic distance estimated from whole genome data (see "Methods")[23,24]. We chose to focus on these clusters, rather than pre-defined clonal complexes, because some clonal complexes, such as CC1 and CC8 were very prevalent in our parental strains, whereas other clonal complexes were represented by only few strains, making it difficult to accurately quantify variability within

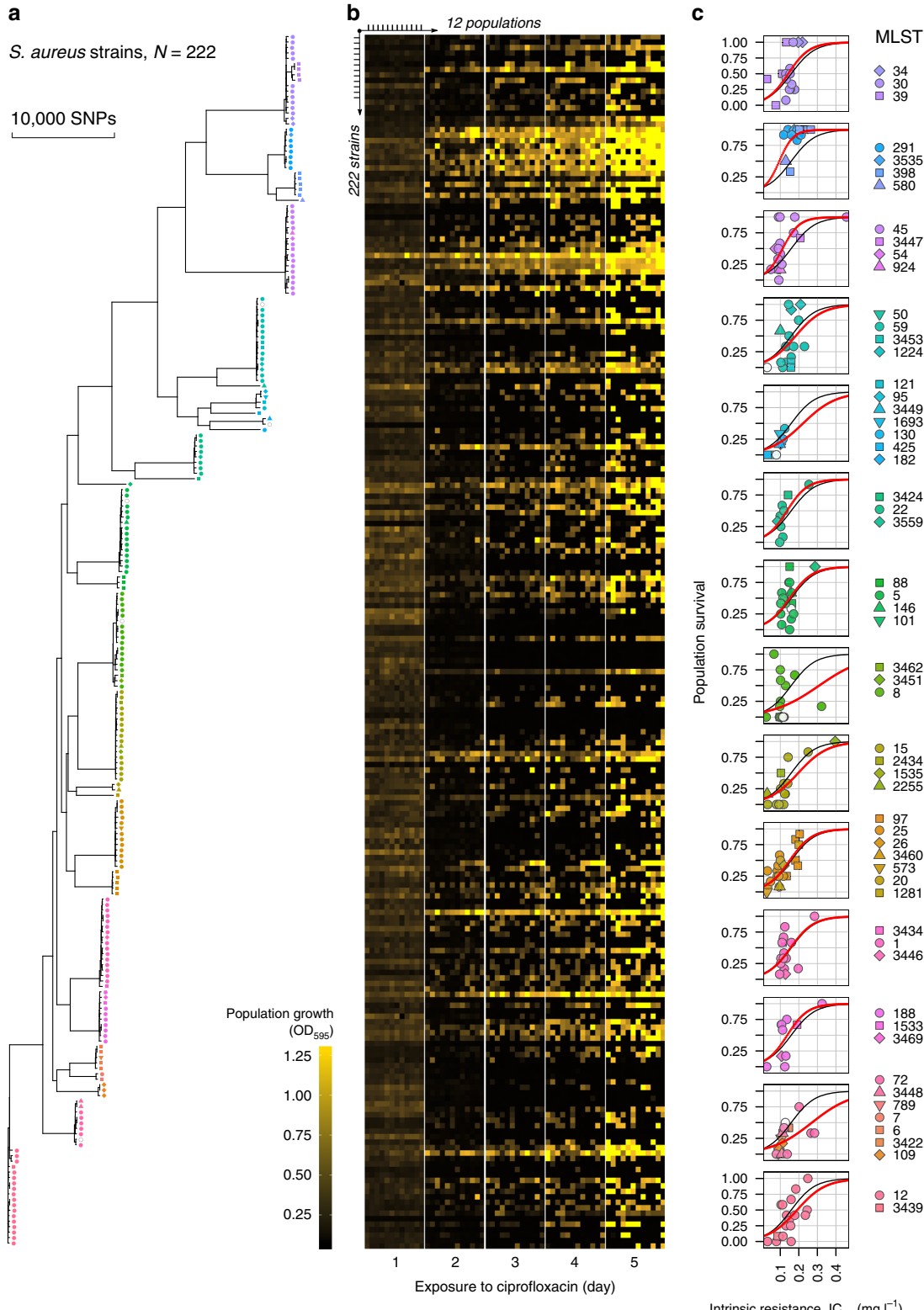

and between clonal complexes (Supplementary Tables 3 and 4). This sampling bias was not deliberate; rather, our pool of parental strains was skewed towards dominant ciprofloxacin-sensitive strains found in the UK.

Evolvability varied significantly between clusters, and among-cluster variation accounted for an additional 16.6% of the variation in evolvability (Fig. 1c, Supplementary Tables 1–3). This effect was largely driven by the fact that clusters with high

levels of intrinsic resistance also had high evolvability (Supplementary Fig. 5). However, we found significant variation in evolvability between clusters after correcting for intrinsic resistance, implying that some strains of S. aureus have elevated evolvability that cannot be explained simply by high intrinsic resistance (likelihood ratio test $\chi^2 = 187.13$, d.f. = 13, $P = 5.94e-33$). For example, strains from cluster 10, which is made up CC398, evolved resistance at a much higher rate than would have

**Fig. 1 Experimental evolution of ciprofloxacin resistance. a** Maximum-likelihood phylogeny of *S. aureus* strains included in this study. The tree was constructed using a whole-genome alignment of 222 strains mapped to the *S. aureus* MRSA252 reference genome and corrected for recombination. **b** Population growth during experimental evolution. Five heatmaps show optical density ($\lambda = 595$) of bacterial populations at the end of each transfer. The optical density varies from low (no growth, $OD_{595} < 0.08$, black) to high density (high growth, $OD_{595} > 1$, bright yellow). In each heatmap, the columns correspond to 12 replicate populations, and the rows correspond to 222 strains. **c** The correlation of population survival and intrinsic resistance in 14 phylogenetic clusters. Each data point represents one strain where population survival is the proportion of replicate populations that evolved resistance, and intrinsic resistance is $IC_{50}$ (half growth inhibition dose) of a parental strain. Colour and shape for each strain indicate MLST (multilocus sequence type). An open circle is shown when MLST could not be determined. 222 strains were assigned to 14 phylogenetic clusters based on a genetic distance matrix obtained from the phylogenetic tree in (**a**). The black curves show a model fit for the logistic regression between survival and intrinsic resistance across all 222 strains (the same for all clusters, GLM: $\chi^2 = 393.93$, d.f. $= 1$, residual d.f. $= 220$, $p < 2.2e{-}16$), and the red curves show cluster-specific effects. Post hoc tests for cluster-specific effects are shown in Supplementary Table 6.

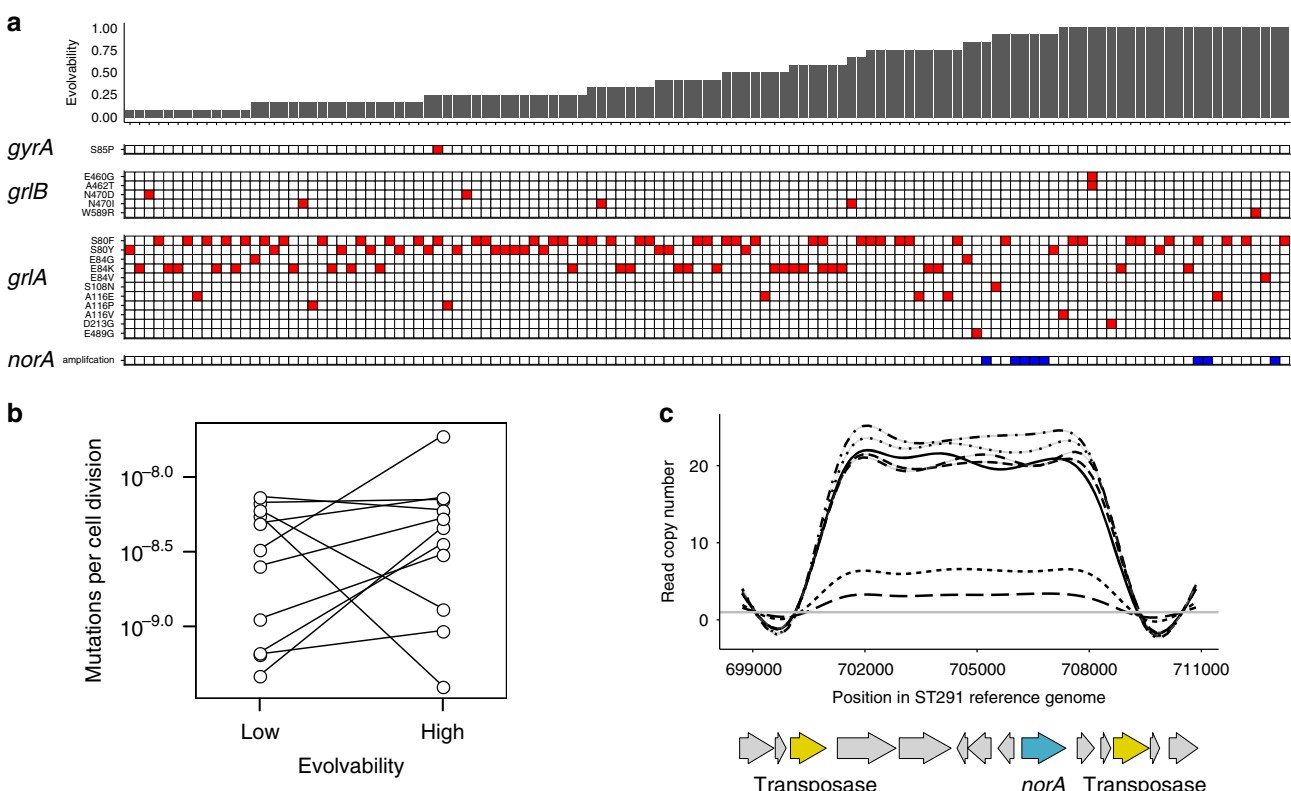

**Fig. 2 Genomic basis of evolved ciprofloxacin resistance. a** Resistance mutations in the evolved *S. aureus* populations identified by whole-genome sequencing ($N = 121$). The panels shows the identified mutations in *gyrA*, *grlB* and *grlA* (red) or the amplification of *norA* gene (blue). The populations are ranked by the evolvability of their parental strains shown as barplots on top of the panel. **b** Mutation rate in high and low evolvability strains. We measured the mutation rate to rifampicin resistance using a Luria-Delbruck fluctuation test. Lines in the figure connect pairs of high and low evolvability strains ($N = 11$ pairs). Two-sided Wilcoxon signed-rank test: $W = 0.8311$, $P = 0.8311$, $N = 12$. **c** Copy number across sites spanning the 7239 bp region that is amplified in ST3535 and ST291 evolved strains. Copy number was calculated by summing the number of reads per site, normalized by the mean sequencing depth across all sites mapping to the ST291 reference genome, and smoothed using a generalized additive model. Gene annotations are shown below: yellow = ISSau1 transposases; blue = *norA*.

been expected based on their intrinsic resistance (Supplementary Table 5, cluster 10, Supplementary Fig. 5).

**Genomic basis of resistance evolution.** One simple hypothesis to explain variation in evolvability between clusters is that some strains might have access to evolutionary paths to ciprofloxacin resistance that are not accessible to others[14,15]. To test this hypothesis, we sequenced the genome of a single evolved population from each of 121 parental strains that spanned the spectrum of evolvability.

The canonical mechanism for *S. aureus* to evolve ciprofloxacin resistance is by point mutations that alter ciprofloxacin targets, including topoisomerase IV (*grlA*, *grlB*) and DNA gyrase (*gyrA*, *gyrB*)[25,26]. Most (106/121; 88%) evolved populations had a single

mutation in these established targets (*grlA*, n = 100; *grlB*, n = 6; Fig. 2a) and the most common SNPs in evolved populations, such as *grlA* E84K and *grlA* S80F/Y, are often found in ciprofloxacin resistant clinical isolates.

We found very few mutations (n = 36; =0.30 mutations per population) outside of these known ciprofloxacin resistance genes, and two lines of evidence suggest that the majority of these mutations were neutral with respect to ciprofloxacin-mediated selection. Strong selection for resistance consistently leads to parallel evolution in key genes involved in antibiotic resistance[14,27–30], but we found no evidence of parallel evolution in genes other than *grlA* and *grlB*. Second, we compared the ratio of replacement to silent mutations in proteins across the genome, under the assumption that positive selection leads to an excess of

replacement mutations that alter protein function relative to what we would expect from spontaneous mutation alone[31]. Replacement mutations ($n = 33$) were more common than silent ($n = 3$) mutations, but the excess of replacement mutations was only marginally greater than the neutral expectation ($K_a/K_s = 2.80$; $N = 36$; $P = 0.095$), suggesting that many non-target site mutations were neutral mutations that hitch-hiked to high frequency with adaptive resistance mutations. The key insight from this analysis is that resistance usually evolved through SNPs in well-defined ciprofloxacin targets, demonstrating a common evolutionary path to resistance in both high and low evolvability strains.

Given the key role of SNPs in resistance, variation in the underlying mutation rate might drive differences in evolvability between strains, as is the case in *M. tuberculosis*[13]. To test this hypothesis, we measured the mutation rate of pairs of high and low evolvability strains sampled across the *S. aureus* phylogeny. Mutation rate varied substantially across strains, but high evolvability was not associated with an elevated spontaneous mutation rate (Fig. 2b, two-sided Wilcoxon signed-rank test; $P = 0.83$, $N = 12$). Furthermore, we found no evidence of an increased substitution rate in high evolvability strains during the selection experiment (Supplementary Fig. 6, W = 273.5, $P = 0.16$, two-sided Mann–Whitney *U* test). Although mutation rate varies between ancestral strains, we found no evidence that high evolvability was associated with hypermutator strains.

One of the most conspicuous cases of high evolvability is the ST291 strains from clonal complex CC398. All eight strains from this group evolved resistance at a high rate, but only 1/8 evolved populations had a classical resistance SNP. Instead, the evolved populations from ST291 lineage carried 3–24-fold amplifications of a ~7 kb region of the chromosome that includes *norA* (Fig. 2c), an efflux pump that contributes to the intrinsic resistance of *S. aureus* towards ciprofloxacin[32,33]. Although the extent of *norA* amplification was impressive, the resistance of evolved populations with *norA* amplification (median MIC = 8 mg l⁻¹; $n = 7$) was only marginally higher than the resistance of populations with topoisomerase substitutions (median MIC = 4 mg l⁻¹; $N = 69$; two-sided Wilcoxon rank sum test $W = 138.5$, $P = 0.0474$).

The amplified region is flanked by two homologous copies of an IS30 family transposase (ISSau1), suggesting that amplification could have been caused by either transposition to different sites in the genome or tandem amplification driven by the homologous copies of ISSau1[34]. To discriminate between these possibilities, we used ISmapper[35] to search for new transposon insertion sites in the chromosome. We did not detect any new ISSau1 insertion sites in the evolved populations, demonstrating that tandem amplification, which can occur at a much higher rate than point mutations[36,37], drove the increase in *norA* copy number. While the ISSAu1 transposon is also found in a few strains outside of CC398 (Supplementary Fig. 7), we speculate that *norA* amplification is not observed in those strains because *norA* is not flanked by nearby copies of ISSau1, as determined by re-analysing the genomes of parental strains.

Given the uniform selective pressure imposed by our experiment, we were surprised by the extent of variation in ISSau1-mediated gene amplification. All of the evolved populations with gene amplification had increased ciprofloxacin MIC, but ciprofloxacin resistance was not correlated with *norA* copy number (Pearson's correlation $r = -0.5733$, $N = 7$, $P = 0.1374$), suggesting that selection for resistance does not explain variation in the extent of *norA* gene amplification. Gene amplification is often associated with fitness costs[37], suggesting that selection to minimize the cost of resistance may drive variation in gene amplification. However, we found no detectable costs of gene amplification, suggesting that selection for high growth rate did

not shape variation in *norA* copy number (Supplementary Fig. 8, two-sided Wilcoxon signed-rank test, $N = 16$, $P = 0.3125$).

In summary, whole genome sequencing allowed us to uncover the mechanistic basis of evolved ciprofloxacin resistance in (114/121 = 94%) of evolved populations. Most strains of *S. aureus* evolved ciprofloxacin resistance via canonical mutations in DNA topoisomerase, albeit at different rates. In contrast, strains from CC398 evolved resistance at a very high rate via a different mechanism involving amplification of the *norA* efflux pump gene mediated by a lineage-specific IS element. We found no evidence of parallel evolution in the remaining seven strains, suggesting that they acquired less common mechanisms of ciprofloxacin resistance (see a list of mutations in Supplementary Data 2).

**Transcriptomic insights into high evolvability**. Sequencing the genomes of evolved populations does not provide any obvious insights into why the rate of evolution of resistance by target alteration varied so widely among non-CC398 strains. One possible solution to this problem is that 'potentiator' genes that alter the fitness effects of topoisomerase mutations may underpin variation in evolvability in non-CC398 strains. To search for candidate potentiator genes, we sequenced the transcriptomes of 14 pairs of high and low evolvability parental strains after exposure to ciprofloxacin. These pairs of parental strains were sampled from different parts of the phylogeny by choosing pairs of strains which are closely related (based on genetic distance) yet showed a striking difference in their evolvability (survival ≥ 9/12 and ≤1/12, Supplementary Table 6). The goal of this paired design was to identify genes that are consistently associated with high evolvability across a range of genetic backgrounds. Gene expression was always measured after 1.5 h of exposure to 1 mg l⁻¹ of ciprofloxacin to match the conditions of the evolution experiment, and our transcriptome experiment did not measure basal gene expression in the absence of ciprofloxacin stress[38].

In total, we identified 179 genes that were differentially expressed between high and low evolvability parental strains (Fig. 3a; Supplementary Data 3). Ciprofloxacin induces the SOS response, which provides both protection against ciprofloxacin and increased mutagenesis[39], making SOS expression an obvious candidate mechanism to potentiate resistance evolution[38]. The SOS regulon was expressed in all strains, but there was no difference in the expression level of SOS regulated genes between high and low evolvability strains (Supplementary Table 7). Instead, overexpressed genes in high evolvability strains were enriched for metabolic functions, while genes that were overexpressed in low evolvability strains were enriched for DNA repair and nucleotide biosynthesis, which is consistent with the DNA damage caused by ciprofloxacin treatment (Supplementary Data 4). The only overexpressed gene in high evolvability strains with a known role in resistance was *norA*, suggesting that this efflux pump might play a very general role in the evolution of ciprofloxacin resistance across *S. aureus* (Fig. 3b).

**Testing the ability of norA to potentiate resistance evolution**. To directly test the role of *norA* expression in evolution, we cloned *norA* into RN4220, a genetically tractable model strain with low evolvability. Overexpressing *norA* from its native promoter marginally increased ciprofloxacin resistance, as we would expect given the established role of this pump in exporting ciprofloxacin from the cytoplasm. However, this effect was very weak, and *norA* overexpression was not actually sufficient to increase the MIC of RN4220 above the clinical break-point concentration of 1 mg l⁻¹ (Fig. 4a, Supplementary Table 8). To test the role of elevated *norA* expression in evolution, we repeated our evolution experiment by passaging 40 replicate cultures of

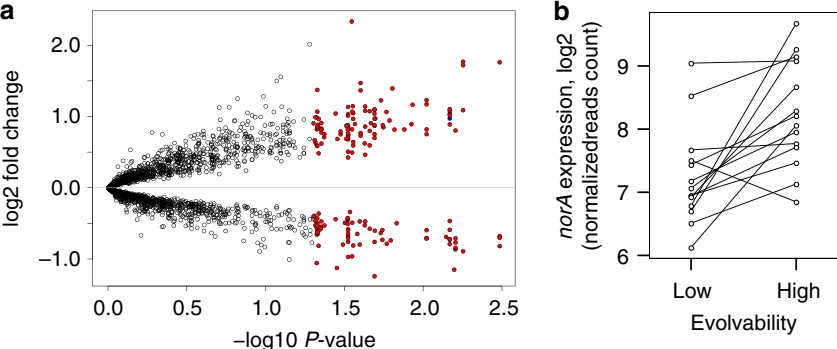

**Fig. 3 Gene expression analysis. a** Plotted points show the average difference (high evolvability−low evolvability) in gene expression between 14 pairs of high and low evolvability strains. Gene expression was measured after 1.5 h of exposure to ciprofloxacin (1 mg l$^{-1}$) and each point represents a gene in the MRSA252 transcriptome (the number of genes is $n = 2047$). Significantly differentially expressed genes are coloured red ($p < 0.05$, two-sided Wald tests) and *norA* is coloured blue. *P*-values were adjusted using Benjamini-Hochberg method. **b** The expression of *norA* in low and high evolvability strains. The read counts for the *norA* gene were normalised by sequencing depth. Pairs of strains are connected with a line (the number of pairs was $N = 14$). Two-sided Wald test: $\log_2$ fold-change = 0.979, standard error = 0.246, $t = 3.979$, adjusted $p = 0.006802703$).

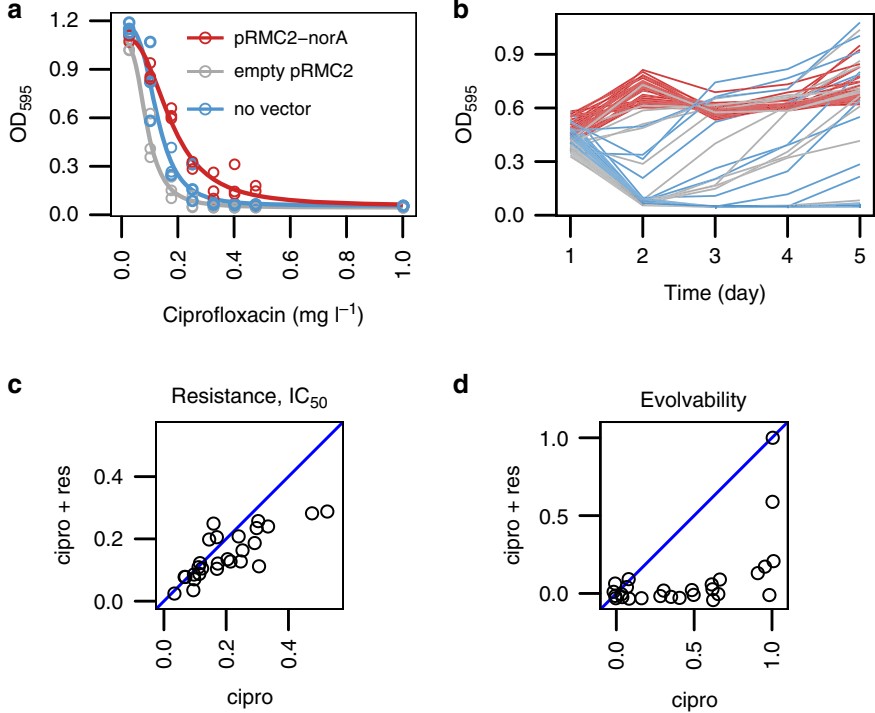

**Fig. 4 The role of *norA* in resistance evolution. a** The effect of *norA* overexpression on ciprofloxacin resistance. Resistance was measured in the RN4220 cells overexpressing *norA* from vector pRMC2 under a native promoter (red), compared to an pRMC2 empty vector control (grey) and a vector-free control (blue). $N = 3$ independent cultures were used per treatment per concentration. Dose-response curves show model fit from the analysis presented in Supplementary Table 8. **b** The effect of *norA* overexpression on the evolution of ciprofloxacin resistance. Optical density ($\lambda = 595$) was measured for five daily transfers with 1 mg l$^{-1}$ of ciprofloxacin in RN4220 cells overexpressing *norA* from the pRMC2-norA vector (red), cells with the empty pRMC2 vector (blue) and the cells without the vector (grey). $N = 40$ independent cultures were used for each type of cells. Statistical analysis in shown in Supplementary Table 10. **c** The effect of reserpine on intrinsic resistance. Intrinsic resistance to ciprofloxacin (IC$_{50}$, mg l$^{-1}$) was determined for a representative set of 27 strains in the presence (*y*-axis) or absence of 33 μM reserpine (*x*-axis) ($N = 5$ per dose/reserpine combination). Two-sided Wilcoxon signed-rank test: $W = 308$, d.f. = 26, $p = 0.003$. **d** Evolvability was determined for the same set of 27 strains with 33 μM reserpine (*y*-axis) or without reserpine (*x*-axis). Evolvability was measured as the probability of population survival after 5 serial transfers at 1 mg l$^{-1}$ of ciprofloxacin ($N = 16$ replicate populations for each strain/reserpine combination). Two-sided Wilcoxon signed-rank test: $W = 226$, $N = 27$, $p$-value = 0.000128.

strain RN4220 in ciprofloxacin supplemented media under basal or elevated levels of *norA* expression. In contrast to the subtle effect of *norA* overexpression on ciprofloxacin resistance, we found that *norA* overexpression had a dramatic effect on population survival under sustained ciprofloxacin treatment, transforming RN4220 from a low evolvability (9/40 populations) to

high evolvability (40/40 populations) strain (Fig. 4b; Supplementary Tables 9, 10).

Despite the recent progress in *S. aureus* forward genetics methods[40], it remains a difficult task to test the importance of *norA* across many *S. aureus* strains. To overcome this problem, we re-assayed the intrinsic resistance of a sub-set of our parental

strains ($N = 27$) in the presence and absence of reserpine, a chemical inhibitor of NorA[41] (Supplementary Fig. 9a, Supplementary Table 11). Reserpine treatment reduced intrinsic resistance to ciprofloxacin (Fig. 4c). Although this impact of reserpine on intrinsic resistance was evident, it translated on average into less than twofold reduction ($IC_{50}/IC_{50 \text{ RES}} = 1.31 \pm 0.47$ s.d.; $MIC/MIC_{RES} = 1.52 \pm 0.49$ s.d.), with a stronger inhibition of strains with high intrinsic resistance (Spearman's correlation $\rho = 0.76$, $p = 7.4\text{e}{-}06$, $n = 27$), suggesting that this higher intrinsic resistance is at least partly based on the efflux of ciprofloxacin.

To understand the evolutionary consequence of *norA* expression more broadly, we repeated our evolution experiment by passaging 16 replicate cultures of these 27 strains in media containing ciprofloxacin, or both ciprofloxacin and reserpine (Supplementary Fig. 9b; Supplementary Tables 12, 13). The overall effect of reserpine was to suppress the evolution of resistance (two-sided Wilcoxon signed-rank test: $W = 226$, $N = 27$, $P = 0.000128$). Reserpine had little effect on low evolvability strains, but it almost completely suppressed the evolution of ciprofloxacin resistance in high evolvability strains, including strains from CC398 (Fig. 4d). Collectively, these results show that *norA* expression modulates the intrinsic resistance of *S. aureus* to ciprofloxacin and potentiates the subsequent evolution of clinically relevant levels of resistance. These findings provide a simple explanation for the positive correlation between intrinsic ciprofloxacin resistance and evolvability that occurs in *S. aureus* (Fig. 1).

**Evolutionary consequences of norA expression**. The most obvious effect of *norA* expression is to provide protection against ciprofloxacin. This protective effect of *norA* could accelerate the evolution of resistance by providing populations with more time to generate resistance mutations until population extinction[42,43]. To test the plausibility of this mechanism, we measured the population dynamics of strain RN4220 under ciprofloxacin treatment (Fig. 5a). As expected, increased *norA* expression reduced the rate of population decline under ciprofloxacin treatment. However, the magnitude of this effect was marginal—*norA* overexpression simply delayed the onset of ciprofloxacin-induced cell mortality by ~1 h. This weak effect of *norA* overexpression on cell viability further highlights the fact that this efflux pump makes a small contribution to the intrinsic resistance of *S. aureus* to high doses of ciprofloxacin (i.e. $1 \text{ mg l}^{-1}$; Fig. 4a). Importantly, it is very difficult to reconcile this marginal effect of *norA* on cell viability with the massive effect of *norA* overexpression on evolvability (Fig. 4b), suggesting that *norA* expression does not accelerate evolution by increasing the rate of appearance of resistance mutations.

The appearance of a ciprofloxacin resistance mutation in a population does not guarantee that a population will successfully evolve resistance, because stochastic processes can lead to the extinction of small populations of resistant mutants, for example during population bottlenecks[44]. All else being equal, mutants with a high fitness upon antibiotic exposure should be more likely to successfully expand their populations and cause populations to evolve elevated resistance. To test the impact of *norA* expression on fitness, we measured the growth rate of RN4220 carrying key ciprofloxacin resistance SNPs under basal and elevated levels of *norA* expression (Supplementary Fig. 10). Elevated *norA* expression did not increase growth rate in wild-type RN4220, which is consistent with our previous ciprofloxacin susceptibility assays (Fig. 4a). In contrast, increased *norA* expression led to large increases in growth rate in ciprofloxacin resistant mutants,

demonstrating positive epistasis between target alteration and antibiotic efflux (Fig. 5b).

*norA* expression could increase the fitness benefit provided by resistance mutations in the presence of ciprofloxacin by reducing the fitness costs of topoisomerase substitutions and/or increasing the protective effect of topoisomerase mutations. To test this hypothesis, we measured the growth of ciprofloxacin resistant mutants under basal and elevated levels of *norA* expression across a range of ciprofloxacin concentrations (Fig. 5c). Elevated *norA* expression did not change the growth rate of resistant mutants in the absence of antibiotics (two-sided paired t-test: $t = 0.42009$, $P = 0.6825$, $N = 12$, Supplementary Fig. 10a), providing further evidence that the expression of this pump has marginal effects on fitness per se (see Fig. 4a and Supplementary Fig. 8). In contrast, we found that elevated *norA* expression dramatically increased the ability of *grlA* mutants to grow at high concentrations of ciprofloxacin (i.e. $1 \text{ mg l}^{-1}$, two-sided paired t-test: $t = -18.712$, $P = 1.09\text{e}{-}09$, $N = 12$). In other words, positive epistasis occurs because ciprofloxacin efflux mediated by *norA* and altered topoisomerase structure (i.e. *grlA*) interact synergistically to increase resistance to high doses of ciprofloxacin without imposing any additional fitness burden.

**Identifying variants associated with high evolvability**. In order to systematically search for genetic variants associated with high evolvability, we carried out a genome-wide association study (GWAS) to test for associations between core-genome SNPs and evolvability across the strains used in this study (Supplementary data 5). The GWAS analysis showed that 16.3% of variation in evolvability can be explained by genetic variability across strains (Supplementary Fig. 11). However, after controlling for multiple testing, the GWAS failed to identify any SNPs that show a significant association with high evolvability.

Given the strong association between *norA* and evolvability, we then focused more closely on variants that are likely to be associated with *norA* expression and function. All of the strains used in this study are predicted to carry a functional copy of *norA* (i.e. no pre-mature stop codons or frameshift mutations). In line with previous work[45], we found that the *norA* coding sequence is polymorphic, and we identified polymorphisms in a previously characterized *norA* promoter region[46–48]. However, the *P* values associated with these polymorphisms are 1000 times higher than the conservative significance threshold used in our GWAS analysis, suggesting that these associations are true negatives. Moreover, we failed to identify any *norA* variants associated with high evolvability through manual inspection of our data.

The expression of *norA* is known to be repressed by the *mgrA* transcription factor[46,49,50], and two lines of evidence from our transcriptome experiment suggest that *mgrA* plays a role in evolvability. First, *mgrA* was overexpressed in low evolvability strains (two-sided Wald test: log2 fold-change = 0.885, standard error = 0.206, $t = -4.305$, adjusted $p = 0.005597$, Supplementary data 3). Second, we found a negative correlation between *mgrA* and *norA* transcript levels across 12 of the 14 pairs of high and low evolvability strains (Supplementary Fig. 12, exact binomial test $P = 0.00091$, $N = 14$), suggesting that *mgrA* represses *norA* in low evolvability strains. Although this relationship between *norA* and *mgrA* is clear at a qualitative level, the quantitative correlation between *mgrA* and *norA* transcript levels is weak (Supplementary Fig. 12), suggesting that other regulatory genes and/or post-translational modification of MgrA play important roles in mediating *norA* expression[49–51]. Interestingly, *mgrA* is highly conserved, and we only identified two polymorphic sites in the strains used in our study. These polymorphisms showed no evidence of association with

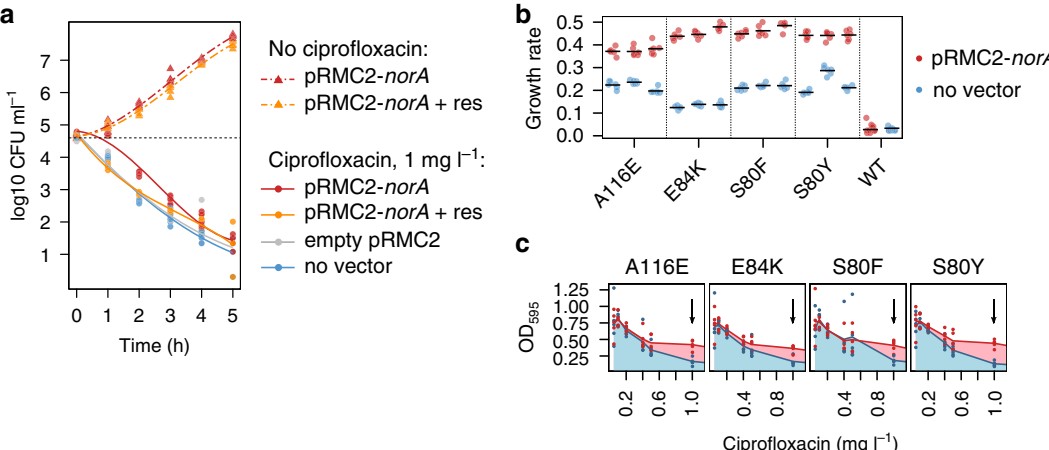

**Fig. 5 The mechanism of norA potentiation. a** Exponentially growing cells were exposed to 1 mg l⁻¹ of ciprofloxacin and viable cells counts were estimated by plating on TSB agar. Four treatments were compared: (i) *norA* overexpression = red solid line, (ii) *norA* overexpression and reserpine (33 μM) inhibition = orange solid line, (iii) empty vector control = grey line, and (iv) no vector control = blue line. In addition, cell density was measured in treatments (i) and (ii) without ciprofloxacin (red dash-dotted line and orange dash-dotted line, correspondingly). For each time-point, six independent replicates per treatment were measured (with the exception of the pRCM-*norA* cells in ciprofloxacin at 0 h which had $N = 5$). The lines show model fit for polynomial regression ($F_{23, 191} = 622.6$, p-value $< 2.2e{-}16$). Post-hoc comparisons are shown in Supplementary Table 14. **b** Growth rate of *grlA* mutants with or without *norA* overexpression. We estimated the growth rate of the RN4220 wild type (WT) and three independently obtained mutants carrying each *grlA* substitution in the presence of 1 mg l⁻¹ of ciprofloxacin. Growth rate in cells carrying pRMC2-*norA* (red) or cells without the vector (blue) was determined using growth curves shown in Supplementary Fig. 10b. Black horizontal lines show the mean growth rate of independent cultures ($N = 6$ per mutant/treatment combination for A116E, E84K, S80F and S80Y, $N = 11$ for WT pRMC2-*norA* and $N = 12$ for WT no vector). Paired two-sided t-test comparing the means in the mutants with and without pRMC2-*norA*: $t = 11.958$, d.f. $= 11$, $p = 1.206e{-}07$. Two-sided t-test comparing pRMC2-*norA* and no vector control in WT: $t = 0.96281$, d.f. $= 12.136$, $p = 0.3544$. **c** Representative dose-response experiments for *grlA* mutants. Red lines show mean optical density with *norA* overexpression, and blue lines show mean density without overexpression. Black arrow shows the concentration used during experimental evolution. Five independent cultures were used per mutant/treatment/concentration. The results for all mutants are shown in Supplementary Fig. 10a. The difference of means between pRMC2-*norA* and no vector cells at 1 mg l⁻¹ for all mutants was tested using two-sided paired t- test: $t = -18.712$, $P = 1.09e{-}09$, d.f. $= 11$.

evolvability in our GWAS ($P$ values $> 10^5$ above conservative threshold), potentially providing evidence that the connection between *mgrA* and evolvability is complex.

## Discussion

We found that the rate and mechanisms of evolution of ciprofloxacin resistance vary dramatically across the diversity of *S. aureus*, an important human pathogen (Figs. 1, 2). Studying the evolution of resistance at this broad scale using a combination of genomic and transcriptomic approaches allowed us to identify lineages with high evolvability and genes that potentiate the evolution of resistance, implying that it is possible to predict the evolution of resistance from genomic data. Remarkably, a single efflux pump gene (*norA*) plays a key role in accelerating the evolution of ciprofloxacin resistance, either by increasing resistance directly, as in the high evolvability CC398 lineage, or by increasing the benefit provided by classical ciprofloxacin resistance mutations that alter DNA topoisomerase.

NorA is a narrow-spectrum efflux pump that is known to contribute to the intrinsic ciprofloxacin resistance in *S. aureus*, but has not previously been linked to clinically relevant levels of resistance[32,41,52,53]. High evolvability strains of *S. aureus* show elevated levels of *norA* expression (Fig. 3), and overexpressing *norA* in a lab strain dramatically increases evolvability (Fig. 4b) by increasing the protective effect of 'classical' ciprofloxacin resistance mutations in *grlA* (Fig. 5b, c). Inhibiting NorA in clinical isolates leads to a marginal loss of intrinsic resistance (Fig. 4c) and a massive reduction in evolvability (Fig. 4d), suggesting that variation in *norA* expression creates a strong association between intrinsic resistance and evolvability across the diversity of *S. aureus* (Fig. 1).

Interestingly, positive epistasis has been found between ciprofloxacin resistance SNPs and mutations that increase drug efflux in Gram-negative bacteria[54–57], suggesting that our findings may help to explain the evolution of ciprofloxacin resistance across a broad spectrum of bacterial pathogens where ciprofloxacin resistance has emerged as an important clinical problem, such as *E. coli* and *P. aeruginosa*. More generally, our findings suggest that more attention should be given to understanding interactions between resistance mutations and 'background' genetic variation that can epistatically modify the fitness effects of canonical resistance mutations[14,30].

Although we have succeeded in linking *norA* expression to evolvability, we were not able to determine the underlying genetic basis of high evolvability outside of CC398. GWAS analysis has promising potential to uncover the genetic basis of bacterial traits, such as high resistance[58] or virulence[24], but this approach was unable to uncover any associations between SNPs and high evolvability. On the one hand, this lack of significance could be explained by the fact that high evolvability is driven by SNPs that are unique to individual lineages of *S. aureus*, in much the same way as lineage-specific IS elements potentiate the rapid evolution of resistance in CC398. Alternatively, it is possible that our analysis simply lacked the statistical power needed to uncover SNPs that are associated with high evolvability, perhaps as a result of limited recombination in *S. aureus*[59]. The negative correlation between the expression of the *mgrA* repressor and evolvability suggests that *mgrA*-mediated repression of *norA* expression helps to constrain evolvability, and an important goal for future work will be to understand the connection between genetic variation, transcriptional regulation and evolvability in pathogenic bacteria.

Our study included a number of strains from clonal complex CC398, containing three sequence types—ST398 ($n = 4$), ST291

($n = 7$) and ST3535 ($n = 1$). The ST398 strains belong to the human-associated lineage of ST398[60] and they evolved via mutations in *grlA*. The ST291 strains are genetically quite different from ST398 and represent a separate lineage within CC398[61]; they are usually associated with humans[62], but more recently isolated from livestock[63,64]. These ST291 strains (and the related ST3535 strain) rapidly and repeatedly evolved in response to ciprofloxacin by ISSau1-mediated tandem amplification of *norA*[34], allowing them to increase ciprofloxacin resistance without any detectable fitness cost. This is perhaps surprising, given that the largest amplifications increased genome size by about 200 kb, which is equivalent to a ~7% increase in genome size. However, a number of other studies have found evidence that *S. aureus* can acquire novel DNA, such as plasmids and SCC*mec* elements, without any additional detectable fitness burden[65,66]. More generally, the high copy number of ISSau1 in CC398 suggests that gene amplification may be an important mechanism of adaptation to novel environments, such as new host species and antimicrobials[67].

Whole-genome sequencing of bacterial pathogens is fundamentally transforming clinical microbiology[68,69], and a key challenge for this field is to exploit the wealth of data provided by genomic sequences to better understand bacterial pathogenesis and epidemiology[3,23,24,70]. By uncovering the link between genotype and evolvability, it should be possible to use pathogen genomic data to help predict the likelihood that resistance will evolve during antibiotic treatment and to alter treatment strategies accordingly. For example, our work suggests that ciprofloxacin should be used cautiously to treat infections caused by CC398 (and in particular ST291), due to the high risk of de novo evolution of resistance. Outside of CC398, measuring the intrinsic ciprofloxacin resistance of clinical isolates could be used in conjunction with phylogenetic data obtained from genome sequencing (Fig. 1) to predict the risk of resistance evolution during treatment. For example, several clusters of strains in Fig. 1c have low evolvability given their intrinsic resistance, making these strains good candidates for ciprofloxacin treatment. Although we have focused on ciprofloxacin resistance in *S. aureus*, it should be possible to use experimental evolution and genomics to predict the risk of resistance evolution for other pathogen/drug combinations. This approach should be particularly useful to predict the evolution of resistance to novel antimicrobials prior to their introduction into clinical use[3,71].

Understanding the genetic basis of evolvability also opens up the possibility of using novel therapies to prevent the evolution of resistance. For example, we have shown that using an efflux pump inhibitor can prevent the evolution of resistance by blocking an important evolutionary path to low costs resistance (see also ref. [14]). Recent work has shown that efflux pumps can also accelerate resistance evolution by increasing the mutation rate[72] or facilitating the acquisition of plasmids[73], suggesting that efflux pump inhibitors may have promising potential to suppress the evolution of resistance[74] for a large number of combinations of pathogen and antibiotic.

## Methods

**Media and reagents.** 96-well flat-bottom plates (Nunc, 260860) and Mueller-Hinton 2 medium (MH2 medium, Sigma-Aldrich, 90922) were used for culturing *S. aureus*, measuring antibiotic resistance and performing all evolution experiments. Tryptic Soy Broth (TSB medium; Sigma-Aldrich, 22092) was used for preparation of competent cells, for selecting transformed clones and for growing *S. aureus* prior to DNA isolation. Incubation was performed at 37 °C and 225 rpm orbital shaking inside a MaxQ 8000 shaking incubator (Thermo Fisher).

Ciprofloxacin (Sigma, 17850) was dissolved in water (5 mg ml$^{-1}$) and stored at −20 °C. The other reagents used in this study were chloramphenicol (25 mg ml$^{-1}$ in ethanol, Acros Organics, 227920250), ampicillin (50 mg ml$^{-1}$ in water, Sigma, A1593), kanamycin sulfate (25 mg ml$^{-1}$ in water, Fisher Chemicals), rifampicin

(50 mg ml$^{-1}$ in DMSO, Millipore, 557303), reserpine (10 mg ml$^{-1}$ in DMSO, Sigma-Aldrich, 83580).

**S. aureus strains.** In total, 222 strains of *S. aureus* were collected from colonized and infected patients in Oxford and Brighton, as previously described[18]. To ensure that these isolates were clonal, the strains were streaked out on TSB agar plates. A single colony was picked to inoculate overnight culture in TSB broth. The overnight cultures of 222 strains were mixed with glycerol to a final concentration of 15% v/v and stored at −80 °C.

**Measuring intrinsic resistance by broth microdilution.** Intrinsic resistance was determined for 222 strains by exposing five replicate cultures per strain to 8 doses of ciprofloxacin (>8800 cultures in total). The strains were handled in five batches with up to 60 strains per batch. Five replicate plates were included for each batch/dose combination, and each plate contained only one replicate of each strain. In order to minimize the effect of well location, five different strain layouts were created by randomization. As a result, every strain was exposed to a given ciprofloxacin concentration in five different plates and at five different well locations. To control for contamination, limit excessive evaporation and avoid an edge effect, 36 wells on the edge of each 96-well plate did not contain bacteria.

The strains were recovered from −80 °C stock by growing overnight cultures for 22 h, diluted 1:100 and randomized across five 96-well plates using a Precision XS automatic pipetting station (BioTek). The resulting master plates were used for transferring 10 μl of bacteria to assay plates containing 190 μl of MH2 broth with ciprofloxacin. The assay plates were placed into the incubator for 22 h. After incubation was completed, optical density was measured at OD$_{595}$ using a Synergy 2 plate reader (BioTek).

OD$_{595}$ values were used to estimate the MIC (minimal inhibitory concentration) and perform a dose-response analysis. A concentration was considered inhibitory if 3/5 replicate cultures did not reach a cut-off value of 0.08 OD$_{595}$ including a blank. The median blank value was 0.042 OD$_{595}$ (s.d. = 0.003), however higher values were occasionally observed due to dust, excessive evaporation, etc. Two assay plates were excluded from the analysis because of unusually high variation in control wells (batch 3, dose = 0.1, replicate plate 5, and batch 4, dose = 0.025, replicate plate 4). In the remaining dataset, only 2/6406 control wells had OD$_{595}$ > 0.08.

For the dose-response analysis, a non-linear 4-parameter model was fitted:

$$f(x) = c + \frac{d - c}{1 + \exp^{[b \times (\log(x) - \log(e))]}},$$

where $c$ is te lower asymptote (i.e. response variable at complete inhibition), $d$ is the upper asymptote (uninhibited growth), $e$ is an inflection point equivalent to a half growth inhibition dose (IC$_{50}$) and $b$ is a slope of the curve at the inflection point. Parameter $c$ was fixed at 0.08 to be consistent with the cut-off value, the remaining 3 parameters were estimated separately for each strain by fitting dose response curves using the drc package in R[75]. The standard errors for IC$_{50}$ were obtained using the coeftest function from the package lmtest The dose response curves were also used to calculate the area under a curve (AUC) using the trapz function in the pracma package.

**Experimental evolution of resistance.** The evolvability of 222 ciprofloxacin-sensitive strains (MIC ≤1 mg l$^{-1}$) was measured by exposing each strain to 1 mg l$^{-1}$ for 5 daily transfers. For each strain, 12 replicate populations were included, amounting to more than 2600 bacterial populations, which were transferred daily in 96-well plates. Prior to the experiment, replicates were distributed across the plates so that each plate had a maximum of one replicate of any strain (except for six plates which had two replicates for a set of 30 strains). For convenience, the strains were handled in sets of up to 60 strains. For each set of strains, three-randomized plate layouts were generated to assign strain locations within a plate. Each randomization layout was used for only four replicate plates. As a result, any given strain had replicate cultures located in 12 plates and in four different well locations (or in six plates for the set of 30 strains mentioned previously). Wells at the plate edges (rows A and H, and columns 1 and 12) were used for blank measurements and as contamination controls.

The strains from −80 °C stock were grown overnight in 200 μl of MH2 (a single culture per strain, five 96-well plates in total). After 22 h of incubation, each plate was used to inoculate three fresh plates by transferring 20 μl of bacteria to 180 μl of MH2. This step was simultaneously used to randomize the locations of strains within plates and was performed using a BioTek Precision XS automatic pipetting station. The resulting 15 plates contained three replicate cultures per strain and had unique strain layouts. After 20 h at 37 °C, the cultures were transferred again to fresh plates with MH2 and returned to the incubator. After the third recovery transfer, 10 μl of culture was inoculated into MH2 medium containing 200 μl of 1 mg l$^{-1}$ of ciprofloxacin. Each culture was used to establish four replicate populations resulting in 12 populations per strain. The plates were incubated for 22 h until the next transfer. In total, five transfers were performed in MH2 medium with ciprofloxacin. At the end of each transfer, the optical density at OD$_{595}$ was measured. By transfer 5, most populations had either gone extinct or evolved resistance and had a high density (more than 90% of surviving populations had an

$OD_{595} > 0.5$). At this point, one more transfer without ciprofloxacin was performed and the evolved populations were frozen in −80 °C in 15% glycerol.

Populations were considered to have survived if they had an $OD_{595} > 0.08$ (including blank) by transfer 5. Only 2/1942 control populations had an optical density above 0.08, indicating a detectable but very low error rate due to experimental artefacts and contamination. In the final dataset, evolvability was defined as the proportion of survived populations per strain, ranging from 0/12 to 12/12.

To determine whether intrinsic resistance is a good predictor for evolvability, a generalized linear model was fitted with a binomial family of distribution of errors. Numbers of survived and extinct populations per strain were used as a response variable (via the logit link function) and $IC_{50}$ as a continuous covariate. In order to control for a lineage effect on evolvability, each strain was assigned to one of 14 phylogenetic clusters. In particular, the cluster package in R was used to perform PAM-clustering algorithm ("partition around medoids", https://cran.r-project.org/package=cluster). The performance of PAM-clustering was checked using the silhouette diagnostic plots provided by the same package. Similar clustering approaches have been used in previous studies to control for a lineage effect[23,24]. Phylogenetic clustering was performed using a genetic distance matrix calculated from a whole genome tree generated from sequence data mapped to MRSA252. Cluster identities and their interaction with the $IC_{50}$-covariate were added to the model as fixed predictors. The emmeans package in R was used to test whether cluster-specific effects differed significantly from the effects calculated for the overall sample.

Ciprofloxacin resistance of evolved populations was determined by broth microdilution method. Evolved populations were selected to maximize phylogenetic diversity and variation in evolvability. Each population was derived from a different parental strain (except strain ERR418532 for which two populations were included). The growth of five replicate cultures for 83 evolved populations was assayed across eight different concentrations of ciprofloxacin, ranging from 0 to 32 mg l$^{-1}$. The assays were performed in two blocks using five-randomized plate layouts. MIC, $IC_{50}$, and AUC was calculated using $OD_{595}$ after 22 h of incubation.

**DNA extraction and sequencing**. DNA was extracted from one representative evolved population for a subset of strains. In this subset, almost all the strains which had an evolvability ≤3/12 ($n = 48$) or ≥9/12 ($n = 44$) were included. Eight strains were excluded because they either failed to grow or the DNA concentration was too low for sequencing. In addition, one evolved population for a representative subset of strains with evolvability ranging from 4/12 to 8/12 ($n = 29$) was included, resulting in a total selection of 121 evolved strains.

Strains were re-grown from −80 °C stock in 2 ml of TSB. Cells were collected by centrifuging for 5 min at 5000 rpm and re-suspended in enzymatic digest buffer (20 mM Tris-HCl, 2 mM EDTA, 1.2% Triton-X100, 0.03 mg ml$^{-1}$ lysostaphin (Sigma-Aldrich, L7386), 25 mg ml$^{-1}$ chicken egg lysozyme (Sigma-Aldrich, 62970)). After 2–4 h of incubation at 37 °C to digest the cell wall, DNA was extracted using DNeasy Blood & Tissue kit (Qiagen) following the manufacturer's protocol for Gram-positive bacteria. The integrity of genomic DNA was checked on 0.9% agarose gel and the concentration was measured using a QuantiFluor dsDNA Kit (Promega).

All parental strains were previously sequenced[76] using an Illumina HiSeq 2000 platform (San Diego, CA, USA) and downloaded from the European Nucleotide Archive Sequence Read Archive (study accession number: PRJEB5261). Evolved strains were sequenced using an Illumina HiSeq 4000 platform with 150 base pair (bp) paired-end reads at the Oxford Genomics Centre (Wellcome Centre for Human Genetics, University of Oxford, Oxford, UK). Sequencing of evolved strains yielded a mean coverage of 302x.

Sequencing reads from both the parental and evolved strains were trimmed using Trimmomatic v0.36[77]. Leading and trailing bases were trimmed if the Phred quality score was less than 20 and a 4-base wide sliding window was used to cut reads when the average quality per base dropped below 15. Any read less than 50 bp long was dropped. Sequences were mapped to the ST36 reference strain MRSA252 (GenBank accession no. BX571856.1) using Stampy v1.0.31[78], with an expected substitution rate of 0.01. Single nucleotide polymorphisms (SNPs) were called using the SAMtools v1.7[79] mpileup command, with the following command line options: -F 0.002 -g -t DP -t SP, and the bcftools v1.7[80] call command with the following command line options: -m -O v -M -P 0.001 -p 0.5. All SNPs supported by fewer than 5 reads or by more than twice the mean sequencing depth (calculated across all sites in the genome) were filtered from the consensus FASTA sequence, which was generated using the bcftools consensus command. Non-unique regions in the reference genome were identified by self-self BLAST analysis of the reference with a word size of 28, using the megablast algorithm in BLAST v2.5.0[81]. Such regions represented 5.93% of the genome and were filtered from the consensus FASTA sequence for each strain. Any base with a heterozygous call under a diploid model was also filtered. Sequences were assembled into de novo contigs for each genome using SPAdes v3.11.1 with the -careful mode turned on[82]. Multi-locus sequence typing was performed on the assemblies using the *S. aureus* MLST website (https://pubmlst.org) sited at the University of Oxford[83].

**Phylogenetic analysis of full parental sequence dataset**. After mapping to the MRSA252 reference genome, genome sequence data were used to construct a maximum likelihood phylogenetic tree in RAxML v8.2.9[84], using the GTRCAT model of nucleotide substitution. Branch lengths were corrected for recombination using ClonalFrameML[85], with a transition/transversion rate ratio (kappa) value of 4.04. This value of kappa was estimated from four independent analyses of subsets of the data ($n = 50$) in PhyML v 3.3.20170105[86].

**Determination of evolved SNPs**. Sequence assemblies for the parental strains were annotated using Prokka 1.12-beta[87]. Snippy v4.0-dev2 (https://github.com/tseemann/snippy) and breseq v0.33.1[88] were used to identify SNPs between each annotated parental genome and the sequence reads from its corresponding evolved genome for 121 strains. Snippy uses bwa to align reads to a reference (here, the assembled parental genome) and FreeBayes to call variants[89,90]. SNPs were required to have a minimum read mapping quality of 60, minimum base quality of 30 and a minimum coverage of 30. A lower mapping coverage threshold (15) was used for SNPs in ancestral isolates due to the lower depth at which these were sequenced. Snippy does not call SNPs at sites with heterozygous genotypes, which here represent diversity within the sequenced culture. This means that any minor variants present in the parental population that subsequently rose to fixation in the evolved population are reported as evolved SNPs by Snippy. As such, only those SNPs reported by both Snippy and breseq, and located at sites without a heterozygous call in the parental strain, were classified as evolved SNPs. In 15/121 strains, no SNPs were identified in genes for established ciprofloxacin resistance targets (topoisomerase IV, DNA gyrase, *norA*). 6/15 strains were ST291/ST3535, for which resistance is inferred to be conferred by amplification of *norA*. Inspection of the variant call format (VCF) file revealed that four of the remaining nine strains had heterozygous calls at known ciprofloxacin resistance sites, suggesting that these evolved populations represented a mix of resistant and susceptible clones. These strains were, therefore, classified as resistant. No known resistance mechanism could be identified in five strains.

**Copy number estimation of amplified region**. Sequence data were remapped to the ST291 reference genome JP80 (GenBank accession no. AP017922.1) to obtain sequence depth estimates for the flanking transposase genes (not present in MRSA252). Copy number was estimated as the total number of good quality reads mapping at each site, normalized by the mean depth across all sites in the mapped genome.

**Evolutionary analysis of norA**. To explore genetic diversity of *norA* within the dataset, BLAST was used to identify the *norA* gene and 250 bp region upstream of *norA* in the sequence assembly for each parental strain. Sequences were aligned using Muscle v3.8.31 and a maximum likelihood phylogenetic tree was constructed in PhyML using an HKY85 model of nucleotide substitution. Tree topology improvement was computed using nearest-neighbour interchange and subtree pruning and regrafting, selecting the best from both searches. This analysis identified three genetically distinct clades within the phylogeny, from which clade-specific consensus sequences were generated by taking the majority allele at each site. All SNPs within the promoter region for each sequence relative to the respective clade-specific consensus and occurring at the tips of the whole genome phylogeny were tested for association with *norA* expression, where this was measured. Ten SNPs were identified in total, five of which were in strains with expression data across all three clades in the *norA* phylogenetic tree.

**Copy number estimation of ISSau1 in parental strains**. Sequence data from all parental strains were mapped to the JP02758_0628 transposase gene located at the 5′ end of the amplified region of the genome using Stampy. Depth of sequence coverage at each site was calculated using the samtools depth program. Transposase copy number was estimated by normalizing the mean sequence depth across all mapped sites by the mean sequence depth across all sites in the corresponding genome mapped to MRSA252.

**Mapping the genomic location of new copies of transposase gene**. The increase in transposase copy number can be a result of tandem amplification or transposition to a different genomic location. To differentiate these two mechanisms, the location of the JP02758_0628 transposase gene was mapped in eight parental and eight evolved strains, in which the transposase amplification was detected. This analysis was carried out using ISmapper v2.0.1[35]. ISMapper takes unmapped paired reads, maps them to an IS element and, then, identifies pairs in which only one read is mapped to the IS element (JP02758_0628). These partially mapped pairs are assumed to be at the junction of a transposon and its genomic neighbourhood, and later used to determine transposition sites within a reference genome (JP80, GenBank accession no. AP017922.1). As a result, a consistent pattern of JP02758_0628 locations was detected between ancestral and evolved sequences, identifying no new transposition sites in the evolved strains and supporting the hypothesis of tandem amplification.

**Genome-wide association study**. A GWAS was carried out using the bugwas[70] package in R[91]. bugwas uses a linear mixed model to test for phenotype associations with both SNPs and lineages, while controlling for relatedness between samples. As input, bugwas requires a sequence alignment, phylogenetic tree and phenotype data for each sample. While it is possible to perform a continuous trait GWAS in bugwas, the evolvability phenotype was not normally distributed and therefore a continuous trait model was not suitable for this dataset. As such, a binary trait GWAS was performed by limiting the analysis to low (evolvability $\leq 2/12$) and high (evolvability $\geq 9/12$) evolvability strains, encoding phenotypes as either 0 or 1 respectively. All strains with intermediate evolvability estimates were excluded from the GWAS analysis, leaving a dataset of 96 strains. A sequence alignment for all strains was obtained by mapping the corresponding sequence data to the MRSA252 reference genome (as described above). A phylogenetic tree was constructed from this sequencing alignment in RAxML v8.2.9[84] using the GTRCAT model of nucleotide substitution. 89,874 biallelic and 8,591 multiallelic SNPs were tested for association with evolvability and annotated in bugwas using the MRSA252 reference genome. P-values were calculated using a likelihood ratio test and a Bonferroni correction was applied.

**Transcriptome analysis of parental strains**. RNA-Seq analysis was performed using the RNA from 30 parental strains. Since differences in gene expression may be confounded by phylogenetic relationship, especially between phylogenetically distant strains, expression was measured for 15 pairs of closely related high and low evolvability strains. First, 15 high evolvability parental strains (evolvability $\geq 9/12$) were chosen to maximize phylogenetic diversity. For each high evolvability strain, the most closely related low evolvability strain (evolvability $\leq 1/12$) was identified (Supplementary Table 6). The strains were grown for 16 h overnight in MH2 broth at 37 °C at 225 rpm, diluted 1:20 in a fresh pre-warmed MH2 and incubated for an additional 1 h and 20 min until they reached a density of 0.160–0.170 OD$_{595}$. Ciprofloxacin was added to achieve a final concentration of 1 mg l$^{-1}$. After 1.5 h, the exposure to ciprofloxacin was stopped by spinning cells for 10 min at 7000 rpm, decanting supernatant and re-suspending the cells in 200 μl of PBS. Finally, 500 μl of RNA-protect Bacteria Reagent (Qiagen) was added and the cells were equilibrated at room temperature for 5 min and then frozen at −80 °C. The cells in RNA-protect solution were thawed, centrifuged for 10 min at 5000 rpm and separated from the supernatant. Next, the cells were mixed with 100 μl of enzymatic digest buffer (30 mM Tris-HCl, 1 mM EDTA, 2 mg l$^{-1}$ protease K (Thermo Fisher), 3.25 mg l$^{-1}$ lysozyme (Sigma-Aldrich, 62970) and 0.15 mg l$^{-1}$ lysostaphin (Sigma-Aldrich, L7386)), re-suspended by pipetting and incubated for 10 min at room temperature with shaking. Three hundred and fifty microliters of RTL buffer (containing 2-mercaptoethanol (Sigma-Aldrich, M3148)) was added and RNA extraction was performed using RNeasy Mini Kit (Qiagen), following the manufacturer's protocol and including the optional step of DNase I on-column digestion. Elution was performed twice using the same volume of 50 μl of RNase-free water. RNA concentration was determined using the QuantiFluor RNA System (Promega). Samples were sent to Oxford Genomics Centre (Wellcome Centre for Human Genetics, University of Oxford, Oxford, UK), where rRNA depletion was performed prior to sequencing using an Illumina HiSeq 4000 platform, generating 75 bp paired-end reads.

Initial principal component analysis identified a likely batch effect, affecting one high evolvability strain. This strain and its low evolvability pair was removed from subsequent analysis. Transcript quantification of the RNA-Seq data from the remaining 14 pairs was performed using Salmon v0.11.3, which maps reads to an indexed reference transcriptome (MRSA252; GenBank accession no. BX571856.1) using a quasi-mapping approach and an inbuilt correction for fragment-level GC bias[92]. Indexing of the reference transcriptome removed sequence-identical duplicate transcripts and used a kmer length of 31 bp. The DESeq2 library in R was used to estimate gene counts and perform differential expression analysis[93] in Bioconductor, with the requirement of at least 1 read mapping to each gene for every strain. Normalised norA transcript counts for individual strains were obtained using the plotCounts function in the DESeq2 library in R.

To test whether genes in different KEGG pathways were up or downregulated in high or low evolvability strains, a KEGG pathway enrichment analysis was carried out using the kegga and topKEGG functions in the limma library in R. A false discovery rate cutoff for differentially expressed genes of 0.05 was used. A given KEGG pathway was considered to be significantly enriched when $P < 0.05$.

**Mutation rate estimation (fluctuation test)**. Mutation rate was determined for 33 parental strains. These 33 strains were selected in the following way. First, 11 high evolvability strains (survival > 10/12) were chosen to maximize phylogenetic representation. Second, each of these strains was paired to the closest low evolvability strain (survival ≤ 1/12). In addition, 11 strains representing CC398 (ST398, ST291 and ST3535) were included. The strains were assigned to three experimental blocks, ensuring a similar representation of high and low evolvability strains in each block.

The strains were re-grown from −80 °C stock in 200 μl of MH2 medium. After 24 h of incubation, strains were subject to a bottleneck by diluting them 10$^7$ times. The bottleneck step was required to remove any pre-existing low frequency mutations. The diluted bacterial cultures were used to establish 16 replicate cultures per strain in 1.2 ml deep well plates (Brand, 701340) with 300 μl of MH2 per well.

After overnight incubation at 37 °C, 5 μl were sampled from eight replicates, diluted 10$^7$ or 10$^8$ times and plated on TSB agar plates to estimate population density. Two hundred microliters of overnight cultures from all 16 replicates per strains were plated on TSB-agar plates containing 100 mg l$^{-1}$ of the antibiotic rifampicin. After 24 h, the TSB plates without antibiotic were photographed using a ColonyDoc-It Imaging Station (UVP, Cambridge, UK). The plates with antibiotic and containing rifampicin-resistant mutants were photographed after 48 h. Colonies were counted in ImageJ[94] using a custom script written in ImageJ macro language.

The colony counts from both types of plates were imported into R. The mutation rate was calculated using the rSalvador package in R, which provides a maximum likelihood estimate of mutation rate under the Luria-Delbrück model and accounts for variation in population density[95]. To compare mutation rates between strains, a Wilcoxon signed rank test (paired, two-sided) was performed.

**The effect of reserpine on ciprofloxacin resistance**. To determine the effect of the norA inhibitor reserpine on intrinsic resistance and evolvability, 27 parental strains (out of 222) were chosen using the following algorithm. First, nine high evolvability strains (survival >10/12) were selected in different parts of the phylogenetic tree. Then, for each high evolvability strain the closest low evolvability strains was identified (survival ≤1/12). In addition, we included nine strains from the high evolvability clonal complex CC398. This set of strains is a subset from the 33 strains used for measuring mutation rates. However, this set of 27 strains is different from the set used in transcriptomic analysis (though, two sets are overlapping), because of different survival thresholds used for defining high and low evolvability strains and because here we included CC398 strains.

The strains were recovered from −80 °C stock and re-grown for two transfers in 200 μl of MH2 broth. During the second transfers, strain layouts were randomized on plates using a Precision XS automated pipetting station (BioTek). Five different randomization layouts were used corresponding to five replicate cultures included per dose/treatment combination. Half of the cultures were exposed to eight different concentrations of ciprofloxacin, and half to the same antibiotic concentration and 33 μM of reserpine. After 22 h of incubation, population growth was measured using a plate reader (OD$_{595}$).

A dose-response analysis was performed by fitting a 4-parameter model. The dose-responses curves were used to estimate IC$_{50}$. The effect of reserpine on IC$_{50}$ was evaluated by using the function EDcomp from the R package drc (version 3.0-1). In addition, IC$_{50}$ was compared using a two-sided Wilcoxon signed-rank test.

**The effect of reserpine on evolvability**. To estimate the effect of efflux pump inhibitor on evolvability, experimental evolution was performed by exposing S. aureus strains to 1 mg l$^{-1}$ of ciprofloxacin, either with or without 33 μM of reserpine. The same 27 strains that were used for measuring the effect of reserpine on intrinsic resistance were used here. For each strain, 16 replicates with reserpine and 16 replicates without reserpine were included. Strain positions on plates were randomized using four different plate layouts. The wells at the plate edges (i.e. rows A and H and columns 1 and 12) were used as controls.

The strains from −80 °C stock were re-grown overnight. During the second transfers, strain positions within the plate were randomized using an automated pipetting station. Half of the populations were challenged with 1 mg l$^{-1}$ of ciprofloxacin, and the other half with 1 mg l$^{-1}$ ciprofloxacin and 33 μM reserpine. Every day, >800 populations were transferred to fresh medium containing the same dose of ciprofloxacin and/or reserpine. Population growth was measured after each transfer by reading optical densities (OD$_{595}$) using a plate reader. The experiment was completed after five transfers.

Due to incomplete solubility of reserpine in water, the optical density of MH2 with reserpine was slightly higher than without reserpine. The average increase in optical density due to reserpine was 0.011 OD$_{595}$, as estimated by comparing a blank sample with and without reserpine (0.049 ± 0.002 and 0.060 ± 0.008, correspondingly; median ± s.d.). Therefore, prior to analysis, measurements from wells containing reserpine were corrected by subtracting 0.011. After applying the correction, populations that reached a density greater than 0.08 OD$_{595}$ were considered to have survived. The difference in survival due to the presence/absence of reserpine was analysed separately for each strain by performing a Fisher's exact test. The resulting p-values were adjusted using Holm–Bonferroni method ($n = 27$).

**Cloning norA into the pRMC2 expression vector and transformation**. Vector pRMC2 was obtained from Addgene (#68940)[96]. The pRMC2 was linearized with EcoRI and KpnI restriction enzymes (NEB) and gel purified using the QIAquick Gel Extraction Kit (Qiagen). The norA gene and a 69 bp upstream region was amplified with PCR, using Phusion DNA Polymerase (NEB). The genomic DNA of strain ERR418607 (ST398) was used as a template. The primers contained 5′-overhangs and were designed for Gibson assembly (see Supplementary Table 15). The linearized vector and the PCR product were assembled using NEBuilder HiFi DNA Assembly Kit (NEB) and transformed into E. coli DC10B (BCCM, Belgium, LMBP 9585). The clones were selected on LB plates with ampicillin (100 mg l$^{-1}$), screened using PCR and confirmed by Sanger sequencing (primers are listed in Supplementary Table 15). The expression vector containing norA was transformed

into *S. aureus* RN4220 (DSMZ, Germany, DSM 26309) by electroporation using the protocol described in ref. [40]. The *S. aureus* transformants were selected on TSB agar plates with chloramphenicol (10 mg l⁻¹) and verified by PCR and Sanger sequencing.

**norA overexpression experiments (resistance and evolvability to ciprofloxacin).** The effect of *norA* overexpression on ciprofloxacin resistance was measured using the broth microdilution method. Although *norA* was cloned under a tetracycline-inducible promoter, increased resistance was also observed without adding the inducer. This is potentially because a native promoter was also cloned as a part of the *norA* upstream region. Further induction of *norA* with an anhydrotetracycline inducer increased resistance but simultaneously affected RN4220 growth rate in the absence of the antibiotic. In order to avoid a negative effect of high overexpression on bacterial fitness and the antimicrobial effect of anhydrotetracycline itself, we decided not use the inducer in all following experiments and relied on the uninduced level of expression.

To assess the effect of *norA* overexpression on resistance, the growth of RN4220 cells carrying pRMC2-*norA* was compared with cells carrying an empty vector and cells without vector. First, *S. aureus* strains were cultured overnight in MH2 (cultures that contained the plasmid were supplemented with 10 mg l⁻¹ chloramphenicol). The next day, cultures were adjusted to a density of $5 \times 10^6$ CFU ml⁻¹ and inoculated into MH2 medium containing 10 different doses of ciprofloxacin. After 22 h of incubation, optical densities were measured. Three replicate cultures were included per dose/genotype combination. The effect of overexpressing *norA* was analysed using a dose-response curve.

To measure the effect of *norA* overexpression on evolvability, the RN4220 cells carrying pRMC2-*norA* or an empty vector pRMC2, and vector-free cells were experimentally evolved for five transfers at 1 mg l⁻¹ of ciprofloxacin. Prior to the experiment, 40 colonies per genotype were selected to establish parallel cultures. After 22 h of incubation in MH2 broth (cultures with the plasmid were supplemented with chloramphenicol). Ten microliters of overnight culture were inoculated to wells containing 190 µl of MH2 and 1 mg l⁻¹ of ciprofloxacin. The cultures were transferred daily to a fresh medium with ciprofloxacin. Populations that reached an optical density of 0.08 or higher by transfer 5 were considered to be alive. To test the differences in evolvability between genotypes, Fisher's exact test was performed on the number of survived/extinct populations and *p*-values were corrected using the Holm-Bonferroni correction.

**Killing assay using ciprofloxacin (the effect of norA expression on survival).** To measure the effect of *norA* expression on cell survival during the exposure to ciprofloxacin, a killing assay was performed including four ciprofloxacin treatments (1 mg l⁻¹) and two control treatments (no ciprofloxacin). The ciprofloxacin treatments were: (i) the RN4220 cells overexpressing *norA*, (ii) the RN4220 cells overexpressing *norA* with 33 µM reserpine (NorA inhibitor), (iii) the RN4220 carrying the empty vector pRMC2 and (iv) RN4220 cells with no vector. The control treatments (no ciprofloxacin) were (v) the RN4220 cells overexpressing *norA* and (vi) the RN4220 cells overexpressing *norA* supplemented with 33 µM reserpine. A day before the experiment, overnight cultures were prepared in MH2 broth including 10 µg ml⁻¹ of chloramphenicol for the bacteria carrying vector pRMC2. The overnight cultures were diluted 1:10 in pre-warmed MH2 broth and incubated for 2.5–3 h (225 rpm, 37 °C) until they entered an exponential growth phase. Next, all strains cells were diluted 1:5 times in pre-warmed MH2, the cell densities were brought to $OD_{595} = 0.072$ in 100 µl of MH2 (including blank). The adjusted cultures were diluted 1:1000 using the pre-warmed media corresponding to their treatments (i.e. with or without 1 mg l⁻¹ ciprofloxacin and with or without 33 µM reserpine) resulting in ~$5 \times 10^4$ CFU ml⁻¹. The cell suspensions were transferred into pre-warmed 96 deep well plates (Brand, 701340) with 250 µl per well, sealed with gas-permeable foil and incubated at 225 rpm at 37 °C. Every hour, six replicate cultures per treatment were sampled, diluted in PBS buffer and plated to MH2 agar plates. For each time point, new cultures were sampled to avoid repeated measurements of the same cultures. The agar plates were incubated at 37 °C for 24 h and, then, photographed using a ColonyDoc-It Imaging Station (UVP, Cambridge, UK). Colonies were counted with ImageJ[94] using a custom script written in ImageJ macro language.

**Obtaining spontaneous grlA mutants.** Two hundred microliters of several independent overnight cultures of the *S. aureus* RN4220 were plated onto MH2 agar plate containing either 1 or 2 mg l⁻¹ ciprofloxacin. Twenty-two colonies were isolated for DNA extraction and Sanger sequencing to identify resistance mutations. The primers used to amplify known fluoroquinolone resistance regions in *grlA*, *gyrA*, *grlB* and *gyrB* genes are listed in Supplementary Table 15. Twenty out of 22 mutants had *grlA* mutation (one A116P, seven A116E, three E84K, four S80Y and five S80F). For further characterization, three independently isolated mutants (i.e., isolated from different overnight cultures) were selected to represent four resistant mutations (A116E, E84K, S80Y and S80F) resulting in total 12 ciprofloxacin-resistant mutants of RN4220.

**Transformation of grlA mutants with pRMC2-norA.** Electro-competent cells were prepared for 12 *grlA* RN4220 following the protocol from[40]. The competent

cells were transformed with the pRMC2-*norA* expression vector. The transformants were selected on TSB-plates supplemented with 10 µg ml⁻¹ of chloramphenicol and confirmed by PCR and Sanger sequencing (primers are listed in Supplementary Table 15).

**Measuring maximum growth rate of grlA mutants.** To estimate the effect of *norA* overexpression on growth rate of the *grlA* mutants, 12 mutants carrying pRMC2-*norA* vector and the 12 corresponding parental mutants having no vector were compared. In addition, the wild-type RN4220 was included as a control (with or without pRMC2-*norA*). The mutants were grown overnight in MH2 broth (containing 10 µg ml⁻¹ of chloramphenicol wherever necessary for pRMC2-*norA* selection). The overnight cultures were diluted 1:1000 in the MH2 broth with 1 mg l⁻¹ of ciprofloxacin and distributed in a 96-well plate with 200 µl of cell suspensions per well. The assay plates were placed into a Synergy 2 plate reader (BioTek) and incubated at 37 °C with continuous shaking. The bacterial growth was recorded by measuring the optical density ($\lambda = 595$ nm) every 10 min for minimum 14 h. For each mutant, six replicate cultures were included, and the experiment was performed in several blocks to accommodate all replicates.

Growth curves were used to estimate the maximum growth rate. A linear regression was used to calculate a slope for each 5 time-point interval of a growth curve in a sliding window fashion (by moving a window one time-point and repeating regression). The resulting distribution of slopes was used to find a maximum growth rate (corresponding to a maximum change in optical density per time unit (an hour)). Because the optical density data were log2-transformed prior the analysis, the obtained estimates should be equivalent to a number of cell divisions per hour (assuming linear relationship between the optical density and cell density during the log growth phase).

**Measuring resistance to ciprofloxacin of grlA mutants.** Twelve *grlA* mutants without vector and their corresponding transformants carrying the pRMC2-*norA* vector were cultured overnight in MH2 broth. Cultures that contained the vector were supplemented with 10 mg l⁻¹ chloramphenicol. The next day, cultures were adjusted to a cell density of $5 \times 10^6$ CFU ml⁻¹ and inoculated into MH2 medium containing 13 different doses of ciprofloxacin (0.01–12 mg ml⁻¹). After 22 h of incubation, optical densities were measured using a plate reader. Five replicate cultures were included per dose/genotype combination. The effect of *norA* overexpression was analysed using a dose-response analysis and, additionally, by performing two-sided paired *t*-test for mean optical densities of mutants obtained at 1 mg l⁻¹ of ciprofloxacin.

**Measuring maximum growth rate of strains with norA amplification.** To estimate the effect of *norA* amplification on fitness, eight parental strains (before amplification) and eight evolved strains (after amplification) were assayed by determining maximum growth rate. These assays were performed in the absence of ciprofloxacin. The parental and evolved strains were grown overnight, diluted 1:1000 in the MH2 broth and inoculated in a 96-well plate (200 µl of cell suspensions per well). The assay plates were incubated inside of a Synergy 2 plate reader (BioTek) at 37 °C with continuous shaking. The bacterial growth was recorded by measuring the optical density ($\lambda = 595$ nm) every 10 min for minimum 14 h. For each strain, three or four replicate cultures were included. Growth curves were used to estimate the maximum growth rate by calculate a slope for each 5 time-point interval of a growth curve in a sliding window fashion. The maximum slope is equivalent to the number of cell divisions per hour during the exponential growth phase in the absence of ciprofloxacin. Maximum growth rate in parental and evolved strains was compared using two-sided paired Wilcoxon sign rank test.

**Statistics and reproducibility.** Experimental evolution of 222 strains was repeated twice. The first attempt failed due to experimental error. Experimental evolution with cell expressing *norA* and experimental evolution with *norA* inhibitor were performed one time. The transcriptome experiment and fluctuation test were performed one time. The determination of resistance to ciprofloxacin in 222 parental strains, in 83 evolved populations, in 27 strains with reserpine, and in 12 *grlA* mutants were performed one time. The experiment with *norA* overexpression in RN4220, the killing assay and estimation of growth rates were performed one time after smaller trial experiments. The results from the trial experiments were reproducible.

**Reporting summary.** Further information on research design is available in the Nature Research Reporting Summary linked to this article.

## Data availability

Whole-genome data for 222 parental strains are available from the European Nucleotide Archive, PRJEB5261. Reference genome sequence of *S. aureus* MRSA252 is available from the GeneBank, BX571856.1. Reference genome sequence of *S. aureus* JP080 is available from the GeneBank, AP017922.1. Whole-genome sequence data of 122 evolved strains were deposited at the Sequence Read Archive, BioProject PRJNA633882. RNA sequencing data for 28 strains are available from Gene Expression Omnibus database, GSE150762.

Other data that support the findings of this study are available from figshare [https://doi.org/10.6084/m9.figshare.c.4984364]. Source data are provided with this paper.

## Code availability

A Jupyter notebook showing the processing of next-generation sequencing data and a script used for counting bacterial colonies are available from Gitlab repository [https://gitlab.com/apson/sa_evolvability/].

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

## Acknowledgements

This project was supported by Wellcome Trust Grant 106918/Z/15/Z. B.Y. is funded by a National Institute for Health Research (NIHR) Clinical Lectureship. We thank K. Foster, D. Wilson and A. San Millan for feedback. We thank the Oxford Genomics Centre at the Wellcome Centre for Human Genetics (funded by Wellcome Trust grant reference 203141/Z/16/Z) for the generation and initial processing of the sequencing data.

## Author contributions

A.P., J.H. and R.C.M. conceived of the study and designed experiments. A.P., N.K. and B.Y. carried out experiments. J.H. and A.P. carried out bioinformatic work and data analysis. R.C.M. and A.P. wrote the paper, which was edited by all authors.

## Competing interests

The authors declare no competing interests.
