## [Peer Review File · Nature Communications]

Reviewers' comments:

Reviewer #1 (Remarks to the Author):

The paper by Papkou et al describes an epistatic interaction between *norA* efflux pump expression and mutations associated with resistance to ciprofloxacin. This is an important paper for the field that reveals an example of an underlying mechanism for the differences between different bacterial clones in their propensity to acquire resistance to antibiotics. It is an extremely well written and presented paper with some beautiful experimental work to support the conclusions made. I have some suggestions for improvement and clarification.

In the title the authors refer to the potential for predicting and preventing the capacity for resistance to emerge. This is an obvious potential application of the findings but as it stands, the title doesn't reflect the research carried out or the conclusions of the study per se. Please modify the title to something more representative of the conclusions of the paper.

norA is well characterised and is deserving of more detail of current understanding in the Introduction or Discussion. Of note, the genetic diversity has been examined previously including allelic variation and identification promoter mutations. The supplemental methods describe *norA* sequence analysis but the data don't seem to be presented anywhere? Have you identified any molecular correlates eg SNPs etc associated with the different levels of *norA* expression in different strains? Is it likely due to differences in promoter efficiency or derepression by regulators eg *mgrA*? Is this tractable by GWAS analysis? Is *norA* present and functional in all strains?

In strains with elevated *norA* expression, is this induced by antibiotic or constitutively higher levels?

As isolates are acquired from a collection obtained in Brighton and Oxford hospitals, do they really 'capture the global diversity of CCs of human-associated *S.aureus*?' If so, how was the global diversity determined? Later, the point is made that strains were skewed towards dominant cipro-sensitive strains found in the UK.

Is there a difference between carriage and bacteraemia isolates with regard to *norA* expression? Could higher *norA* expression levels reflect adaptation to a hospital setting?

Is the amplification of the *norA* region in ST398 strains a tandem amplification or does the region become circularised/extrachromosomal? Previously high level oxacillin resistance was reported to be due to tandem amplification of SCCmec (Gallagher et al AAC 2017).

RN4220 is heavily chemically mutagenized and very far removed from anything found in an infection, clinic or human nose. What was the rationale for using this strain?

The authors state that only a few model strains are genetically tractable- I don't think that statement is now correct- there are now efficient tools for circumventing restriction barriers that allow most strains to be manipulated.

There are 2 major clades of CC398, one which circulates in human populations and the other which is livestock-associated and causes zoonotic infections of humans. Which do your human isolates belong to? This should be mentioned at the relevant point of the discussion, particularly with regard to the 'threat posed'.

Final paragraph of the discussion- the word 'incredible' should be deleted.

Reviewer #2 (Remarks to the Author):

This is one of the widest studies so far published analyzing the effect of the genomic background in the selection of antibiotic resistance. The methods are strong and generally support the conclusions. Nevertheless, there are some aspects that require further clarification and some previous works in the field should be discussed in more detail.

Concerning fitness, it seems that the authors mix up fitness in the presence and in the absence of antibiotics. For instance, I guess that the sentence at the beginning of page two refers to fitness in the presence of quinolones, there is nothing wrong in using fitness when antibiotics are present, but it must be stated: "fitness in the presence of antibiotics".

Also on this page, the authors highlight that genetic diversity plays a key role in shaping resistance evolution. Although it is true that there are relevant differences in the evolvability, one thing that may also be important to highlight is just the opposite: despite this variability, only two elements contribute to resistance: *norA* and topoisomerases. This means that evolution trajectories are qualitatively similar and quantitatively different. In other words, the mechanisms (question iii) are very similar, but the rate differs.

The main cause of increased evolvability is *norA* level of expression. However, the reasons for the differential expression in the strains that do not contain the amplification are not discussed. Do the authors have any insight into the mechanisms dealing with *norA* increased expression?

Page 5 Line 4: Please comment either here or after discussing the information of Fig S4 the levels of *norA* expression in these two sets of strains.

Page 6, line 8. Please state the method, I guess it is MLST, but could also be WGS and a clarification would be welcomed.

Page 8: Line 23. The absence of ISSAu1 on other strains comes from the analysis of the strains here sequenced or comes from previous studies? Please clarify

Page 9. Page 9, line 16. A brief discussion of the mutations contained by these strains will be of interest to others working in the field and, once the information is available, can be easily included.

Page 10, lines 12-18. I am not sure if these data refer to the expression on the absence/presence of antibiotics of the rate of induction by the quinolone. Please clarify and, if the information is in the presence of quinolone, discuss expression without antibiotics as well. Maybe DNA repair is not properly working in the high evolvability strains? Any insight on the mechanism of *mgrA* and *norA* different levels of expression in the high and low evolvable strains?

Page 14. Lines 6-11. Excepting CC398, it is not clear that lineages with high evolvability can be identified and the same happens with genes when genomic data are analyzed. The only gene that seems to be relevant is *norA*, ubiquitously present in *S. aureus*, and its relevance does not depend on its presence/absence (genomics), but in its expression level (transcriptomics).

The authors would like to take into consideration previous works showing the existence of epistatic interactions between mutations increasing the expression of efflux pumps and mutations in genes encoding the bacterial topoisomerases. See for instance Antimicrob Agents Chemother. 2012 or PLoS Pathog. 2009 Aug;5(8):e1000541.

The authors would also like discussing former pieces of evidence showing that the inhibition of

efflux pumps reduces the chances of acquiring antibiotic resistance. See for instance *Antimicrob Agents Chemother.* 2001 Jan;45(1):105-16, where it is stated "In the case of levofloxacin alone, the frequency was approximately 10^{-7} CFU/ml. In contrast, with an EPI, the frequency was below the level of detection ($<10^{-11}$). In summary, we have demonstrated that inhibition of efflux pumps (i) decreased the level of intrinsic resistance significantly, (ii) reversed acquired resistance, and (iii) resulted in a decreased frequency of emergence of *P. aeruginosa* strains that are highly resistant to fluoroquinolones." These results are in line with the findings of the current article and should be discussed.

Reviewer #3 (Remarks to the Author):

The authors claim to show that

--Different strains of pathogenic *S. aureus* differ widely in their rate of evolving quinolone-resistant topoisomerase

--NorA (quinolone efflux pump) amplifies fitness benefit of topoisomerase mutation.

The results are very interesting, although previously it is known that drug pump AcrAB predisposes pathogens to drug resistance (Nolivos) as do tandem amplifications (Andersson; Nicoloff). The discussion of epistasis is very interesting but needs to be clarified.

Overall, the authors need to clarify their explanation of experimental design, as indicated below.

They should discuss the role of small fitness differences that cannot be measured by MIC.

4, 18-20 How many passages and generations (doublings) occurred before resistance? Perhaps some of the supplemental figures belong in main manuscript.

5, 3-4 Clarify definition of "intrinsic resistance".

5, 13 What is meant by "susceptibility"? Here "intrinsic resistance" is defined partly, but after the term was already introduced above. Fig S4 should be in main paper.

Line 23 seems to contradict lines 17, 18.

7, 5 Assume "from each of 121 parental strains" is meant.

7, 14-20 The explanation for non-selection of not-target gene SNPs is good but clarify the claim regarding silent versus replacement mutations. What do all the numbers represent? What numbers would be expected otherwise, if the non-target alleles had selective value?

8, 4 Mutation rate or mutation frequency? Mutation rate is notoriously difficult to measure.

8, 13 Now we learn that *norA* is a major part of intrinsic resistance. This information needs to come earlier, at the definition of intrinsic resistance, not postponed like a mystery novel.

9, 5-10 There is much previous evidence that resistance gene amplification (Andersson; Nicoloff heteroresistance) lead to drug-resistant pathogens. It is likely that highly variable copy number would not quantitatively parallel the degrees of drug resistance. As for "detectable costs of gene amplification," there is evidence in other systems of high costs of gene amplification, both due to the translation costs and the costs of function of the expressed product. The authors need to explain how they set up their fitness assay, not just show a figure.

Note that MIC is a poor indicator of relative fitness. Coculture competitions are required to demonstrate relative fitness differences at levels of drug 1/100 that of MIC.

17, 8-10 Microplate evolution protocols are subject to cross-transfer between wells. The authors should address how they avoided crossover, and whether any evidence of crossover was seen in their results.

Response to reviews

Reviewers comments are shown in italics (we have numbered their points to make it easier to link responses to different reviewers)

Our responses to reviews are shown offset

Reviewer #1 (Remarks to the Author):

The paper by Papkou et al describes an epistatic interaction between norA efflux pump expression and mutations associated with resistance to ciprofloxacin. This is an important paper for the field that reveals an example of an underlying mechanism for the differences between different bacterial clones in their propensity to acquire resistance to antibiotics. It is an extremely well written and presented paper with some beautiful experimental work to support the conclusions made. I have some suggestions for improvement and clarification.

1.1 In the title the authors refer to the potential for predicting and preventing the capacity for resistance to emerge. This is an obvious potential application of the findings but as it stands, the title doesn't reflect the research carried out or the conclusions of the study per se. Please modify the title to something more representative of the conclusions of the paper.

We agree with the reviewer that the title was not representative of the conclusions of the study and we have changed the title to “Efflux pump activity potentiates the evolution of antibiotic resistance across *S. aureus*”

1.2 norA is well characterised and is deserving of more detail of current understanding in the Introduction or Discussion. Of note, the genetic diversity has been examined previously including allelic variation and identification promoter mutations.

We agree with the reviewer that the previous version of this paper did not give an adequate overview of previous work on *norA* diversity and regulation. We have now cited previous work examining *norA* genetic diversity (references 45; page 16, line 1-4) and regulation (references 46-52, page 16, lines 8-21).

1.3 The supplemental methods describe norA sequence analysis but the data don't seem to be presented anywhere? Have you identified any molecular correlates eg SNPs etc associated with the different levels of norA expression in different strains ? is it likely due to differences in promoter efficiency or derepression by regulators eg mgrA Is this tractable by GWAS analysis ?

Is norA present and functional in all strains ?

In response to this point, we have included an important new section in the manuscript “Identifying variants associated with high evolvability” (pages 15-16). We systematically searched for SNPs associated with high evolvability using a GWAS analysis (Supplementary Figure 11, Supplementary Data 4), but this approach failed to identify any significant associations between SNPs and evolvability.

All strains used in our study are predicted to carry functional copies of *norA*, and our GWAS analysis included polymorphisms in the *norA* promoter and coding sequence. The P values for all *norA* related SNPs were >100x above the significance threshold used by our GWAS, suggesting that these associations were true negatives.

norA expression has previously been shown to be repressed by the *mgrA* transcription factor (page 16, lines 8-10, references 46, 49, 50). Our transcriptome analysis found a limited correlation between *mgrA* expression levels and evolvability and *norA* expression (Supplementary Figure 12). These results suggest that *mgrA* may influence evolvability via *norA* repression. However, *mgrA* is very highly conserved (only 2 homoplasic SNPs), suggesting that polymorphisms elsewhere in the genome shape *mgrA* expression level. These findings are presented in the results section (page 15, lines 14-20 ; page 16 line 8-21) and further discussed in the conclusion (page 18, line 6-19).

1.4 In strains with elevated norA expression, is this induced by antibiotic or constitutively higher levels?

Differential gene expression was determined by comparing high and low evolvability strains **after** the exposure to ciprofloxacin, as now emphasized in the revised manuscript (page 11, line 6-8). The ciprofloxacin concentration and other conditions were closely matched to the conditions during experimental evolution, so that we could make use of transcriptome data to explain different evolutionary outcomes. We used 30 different strains for our transcriptome study. Unfortunately for economical and logistical reasons, it would have been burdensome for us to include a control group, as it would have required 30 more samples (+5000£). Based on the data we have, we can only say that gene expression was different after the exposure to ciprofloxacin. Now we emphasise it more clearly in the main text (page 11, line 6-8).

1.5 As isolates are acquired from a collection obtained in Brighton and Oxford hospitals, do they really 'capture the global diversity of CCs of human-associated S.aureus'? If so, how was the global diversity determined? Later, the point is made that strains were skewed towards dominant cipro-sensitive strains found in the UK.

We agree with Reviewer #1 comment and change 'capture the global diversity' to a 'capture a diverse set of clonal-complexes of human-associated S. aureus' (page 3, line 20).

1.6 Is there a difference between carriage and bacteraemia isolates with regard to norA expression? Could higher norA expression levels reflect adaptation to a hospital setting?

The majority of the isolates used in our study were collected from blood stream infections (reference 18). Although it would be very interesting to try to correlate *norA* expression level to infection status, our collection of strains would not be ideal for this kind of analysis due to the bias towards isolates from infections and the fact that we only measured *norA* expression levels in 30 strains. We agree with the reviewer that this is a very interesting idea, but this goes beyond the scope of the present study.

1.7 Is the amplification of the norA region in ST398 strains a tandem amplification or does the region become circularised/extrachromosomal? Previously high level oxacillin resistance was reported to be due to tandem amplification of SCCmec (Gallagher et al AAC 2017).

This is a very interesting point, and we thank the reviewer for bringing up this recent study. In this case, the copy number of this region could have increased due to transposition (i.e. the region between the two copies of ISSau1 forming a composite transposon) or tandem amplification. To discriminate between these possibilities, we checked for novel ISSau1 insertion points using ISmapper, which has been developed to map transposon movement using re-sequencing data (such as during experimental evolution, see reference 35). Using this method, we did not detect novel ISSau1 insertion sites in the evolved samples compare to their parental strains, providing good evidence for tandem amplification, as in the cited paper. We have added this point to the revised manuscript (page 9, lines 9-18), and we have added the citations to the Gallagher *et al.* study (reference 34).

1.8 RN4220 is heavily chemically mutagenized and very far removed from anything found in an infection, clinic or human nose. What was the rationale for using this strain? The authors state that only a few model strains are genetically tractable- I don't think that statement is now correct- there are now efficient tools for circumventing restriction barriers that allow most strains to be manipulated.

We used RN4220 for two main reasons: (i) it is the easiest to manipulate genetically and (ii) it is sensitive to ciprofloxacin (unlike some other commonly used strains, for example MRSA252 and USA300). We acknowledge that there has been significant progress in “breaking through a transformation barrier” in other *S. aureus* strains, however the improved protocols still require micrograms of input plasmid, and RN4220 is still the easiest to be manipulated. In addition, one of our experiments included many independent transformations (i.e., the transformation of multiple *glrA* mutants), therefore we compromised in favour of higher efficiency.

In the revised version of the manuscript, we acknowledge the progress that has been made in developing more tractable strains for *S. aureus* genetics (reference 40, p 12, line 14-15).

1.9 There are 2 major clades of CC398, one which circulates in human populations and the other which is livestock-associated and causes zoonotic infections of humans. Which do your human isolates belong to ? This should be mentioned at the relevant point of the discussion, particularly with regard to the 'threat posed'.

In our dataset, there are a number of sequence types (MLSTs) from clonal complex CC398. The most studied sequence type is ST398 which includes human and livestock associated lineages. We only had 4 strains from ST398 (all from human lineage). All the other CC398 strain in our sample were identified as ST291 (n=7) and a closely related ST3535 (n=1). Based on whole genome data in a previous study (reference 62), ST291 was separated into a distinct lineage within CC398 (quite different from human and livestock-associated ST398). This lineage has been less studied in comparison to ST398, and it has been isolated from human infections, as well as from livestock (see references 63-65).

In the revised manuscript, we clarify that *norA* amplification happened only in ST291 and ST3535 (page 18, line 20 - page 19, line 4).

1.10 *Final paragraph of the discussion- the word 'incredible' should be deleted.*

We removed the word 'incredible'

Reviewer #2 (Remarks to the Author):

This is one of the widest studies so far published analyzing the effect of the genomic background in the selection of antibiotic resistance. The methods are strong and generally support the conclusions. Nevertheless, there are some aspects that require further clarification and some previous works in the field should be discussed in more detail.

2.1 Concerning fitness, it seems that the authors mix up fitness in the presence and in the absence of antibiotics. For instance, I guess that the sentence at the beginning of page two refers to fitness in the presence of quinolones, there is nothing wrong in using fitness when antibiotics are present, but it must be stated: "fitness in the presence of antibiotics".

We agree with the reviewer that this point could have been made more clearly, and we have modified the following sentences (page 7 line 21, page 14 line 12) and changed the abstract (page 2, line 2) by adding "under ciprofloxacin treatment"

*2.2 Also on this page [the abstract], the authors highlight that genetic diversity plays a key role in shaping resistance evolution. Although it is true that there are relevant differences in the evolvability, one thing that may also be important to highlight is just the opposite: despite this variability, only two elements contribute to resistance: *norA* and topoisomerases. This means that evolution trajectories are qualitatively similar and quantitatively different. In other words, the mechanisms (question iii) are very similar, but the rate differs.*

This is an interesting point – it is not clear if our default expectation should be that all strains will follow the same evolutionary trajectory, or that each strain will have its own unique trajectory. The reviewer suggests that only two elements contribute to resistance: *norA* and topoisomerases. It is clear the two mechanisms of resistance are topoisomerase substitutions and *norA* amplification (in CC398), as emphasized in the abstract. However, we do not fully understand the genetic basis of variation in *norA* expression. Given this, we would like to keep the focus of the abstract on the importance of diversity in resistance.

*2.3 The main cause of increased evolvability is *norA* level of expression. However, the reasons for the differential expression in the strains that do not contain the amplification are not discussed. Do the authors have any insight into the mechanisms dealing with *norA* increased expression?*

This point is very similar to 1.3, and we will briefly cap our response to this point here. It is clear from our transcriptome analysis that variation in the expression of *norA* and, potentially, of *mgrA* plays an important role in determining evolvability

(Figure 3 B, new Supplementary Figure 12). However, the genetic variation that underpins this variation is unclear – GWAS failed to identify any SNPs associated with evolvability (new Supplementary figure 11), and there is no evidence that polymorphisms in *norA* and *mgrA* were associated with increased evolvability. *mgrA* is known to be regulated by other global regulatory proteins and MgrA undergoes post-translational modification, suggesting that the genetic architecture of evolvability may be complex. This issue is addressed in the new results section “Identifying variants associated with high evolvability” (page 15-16) and in the conclusions (page 18, line 6-19) and a more extensive discussion is provided in our answer to 1.3.

2.4 Page 5 Line 4: Please comment either here or after discussing the information of Fig S4 the levels of norA expression in these two sets of strains.

Reviewer #2 suggests bringing the information on the differences in *norA* expression into the discussion of results from the experimental evolution. After carefully considering this suggestion, we still think it is more appropriate to keep a chronological order. First, experimental evolution was the first key experiment in our study and, therefore, deserved to be fully covered before bringing in all the following experiments. Second, the design of transcriptome study itself was based on the result of experimental evolution, therefore it might be confusing to start describing the transcriptome analysis before presenting the results of experimental evolution. Third, the transcriptome evidence for the role of *norA* would not be as strong if we had not presented other pieces of evidence: (i) correlation between intrinsic resistance and evolvability, (ii) no difference in mutation rate between high and low evolvability strains, and (iii) the same genetic mechanism of resistance in all strains except for one lineage. Finally, we mention the efflux pump in the title and in the abstract, so it should not come as a surprise.

2.5 Page 6, line 8. Please state the method, I guess it is MLST, but could also be WGS and a clarification would be welcomed.

We changed the text to clarify that grouping into phylogenetic clusters was based on whole genome sequence data (page 6, line 12)

2.6 Page 8: Line 23. The absence of ISSAu1 on other strains comes from the analysis of the strains here sequenced or comes from previous studies? Please clarify

The presence/absence and copy number of ISSAu1 was determined by re-analysing Illumina reads for all parental strains available from a previously published study (ref 18). This point is clarified in the revised manuscript: “as determined by re-analysing the genomes of parental strains” (page 9, line 18)

2.7 Page 9. Page 9, line 16. A brief discussion of the mutations contained by these strains will be of interest to others working in the field and, once the information is available, can be

easily included.

Thank you for this suggestion. We have now included a list of mutations in Supplementary data 2 including non-target mutations. However, we do not discuss these mutations in the text as it is unclear to what extent these mutations are linked to elevated ciprofloxacin resistance or fitness.

2.8 Page 10, lines 12-18. I am not sure if these data refer to the expression on the absence/presence of antibiotics of the rate of induction by the quinolone. Please clarify and, if the information is in the presence of quinolone, discuss expression without antibiotics as well.

This point is very similar to 1.4, and we briefly re-cap our answer here. We measured gene expression in the presence of ciprofloxacin, as we were trying to understand how gene expression impacted evolvability under the conditions of our evolution experiment. This is now highlighted in the revised manuscript (page 11, line 4-8). We measured the transcriptomes of 30 strains, and including measurements in the absence of antibiotics would have been associated with a large extra cost (circa 5000€) and workload.

2.9 *Maybe DNA repair is not properly working in the high evolvability strains?*

We took two approaches to addressing this problem. First, we measured mutation rates in the presence of ciprofloxacin, and we found no difference in mutation rate between high and low evolvability strains (Figure 2B, page 8, lines 13-18). Second, we compared the number of mutations that high and low evolvability populations acquired during the experiment. Again, we found no difference in rates of molecular evolution (page 8, lines 18-20), providing further evidence that differences in DNA repair and mutagenesis do not explain variation in evolvability.

2.10 *Any insight on the mechanism of mgrA and norA different levels of expression in the high and low evolvable strains?*

Please see the response to 2.3.

2.11 *Page 14. Lines 6-11. Excepting CC398, it is not clear that lineages with high evolvability can be identified and the same happens with genes when genomic data are analyzed. The only gene that seems to be relevant is norA, ubiquitously present in S. aureus, and its relevance does not depend on its presence/absence (genomics), but in its expression level (transcriptomics).*

Our analysis shows that 16% of variation in evolvability can be accounted for by phylogenetic information (page 6, line 19-20). Although cluster 10 (which includes CC398) contributes to this pattern, it is clear that there is variation in evolvability between other clusters (see Figure 1C); for example, 2 of the clusters tend to have low evolvability, making strains from these clusters good candidates for ciprofloxacin treatment. Given this variation, whole genome sequence data can contribute to predicting the evolvability of strains, and we have clarified this in the revised manuscript (page 19, line 19-22).

2.12 The authors would like to take into consideration previous works showing the existence of epistatic interactions between mutations increasing the expression of efflux pumps and mutations in genes encoding the bacterial topoisomerases. See for instance *Antimicrob Agents Chemother.* 2012 or *PLoS Pathog.* 2009 Aug;5(8):e1000541.

The authors would also like discussing former pieces of evidence showing that the inhibition of efflux pumps reduces the chances of acquiring antibiotic resistance. See for instance *Antimicrob Agents Chemother.* 2001 Jan;45(1):105-16, where it is stated "In the case of levofloxacin alone, the frequency was approximately 10^{-7} CFU/ml. In contrast, with an EPI, the frequency was below the level of detection ($<10^{-11}$). In summary, we have demonstrated that inhibition of efflux pumps (i) decreased the level of intrinsic resistance significantly, (ii) reversed acquired resistance, and (iii) resulted in a decreased frequency of emergence of *P. aeruginosa* strains that are highly resistant to fluoroquinolones." These results are in line with the findings of the current article and should be discussed.

We are grateful to the Reviewer #2 for bringing this additional literature. We have now included it to our discussion (references 54-57, page 17 line 20 – page 18, line 5).

Reviewer #3 (Remarks to the Author):

The authors claim to show that

--Different strains of pathogenic S. aureus differ widely in their rate of evolving quinolone-resistant topoisomerase

--NorA (quinolone efflux pump) amplifies fitness benefit of topoisomerase mutation.

3.1 *The results are very interesting, although previously it is known that drug pump AcrAB predisposes pathogens to drug resistance (Nolivos) as do tandem amplifications (Andersson; Nicoloff). The discussion of epistasis is very interesting but needs to be clarified.*

It is unclear from this comment how we should clarify our discussion of epistasis. We have included citations to previous studies that have highlighted the role of efflux pumps (references 54-57, 73-74) and tandem amplification (references 34, 36-37) in the paper.

Overall, the authors need to clarify their explanation of experimental design, as indicated below.

3.2 *They should discuss the role of small fitness differences that cannot be measured by MIC.*

We completely agree with the reviewer that standard MIC assays are not able to capture subtle and quantitative variation in resistance. For this very reason, we used IC50, which is a continuous variable and a lot more sensitive than MIC.

We also would like to point out that we did our best to maximise accuracy by including 5 technical replicates across a fine range of concentrations (5*8=40 data points per single IC50 estimate), randomising strains' location in mutliwell plates,

and estimating IC50 using a statistical framework (dose-response analysis). This approach to analysing resistance is explained in the methods (page 22, lines 5-21) and in more details in Supplementary methods. In general, the fact that we obtained a strong correlation between IC50 and evolvability suggests that our method was good enough to capture this quantitative variation across strains.

3.3 4, 18-20 How many passages and generations (doublings) occurred before resistance? Perhaps some of the supplemental figures belong in main manuscript.

This is an interesting point. We show the OD data in Figure 1B, and replot this data in Supplementary Figure 1. It is clear from this data that different populations that evolve resistance do so at different times, and the divergence between populations in OD increases over time. In most experimental evolution studies (where replicate populations grow by a fixed factor every day) it is easy to calculate the number of population doublings that have occurred during an experiment (for example, see a classic paper by Lenski et al. published in 1991 *Am. Nat.* 138:1315-1341). In this case, however, the number of population doublings is not easy to calculate. As an extreme example, most populations actually became smaller during the first day (Supplementary Figure 1; Figure 4A). Subsequent population rebounds occurred at different days in the experiment in different strains/replicates. For this reason, it is not easy to accurately estimate the length of this experiment in terms of generations, and the length of the experiment in terms of generations is not crucial to any of the conclusions of this study.

3.4 5, 3-4 Clarify definition of “intrinsic resistance”.

We thank the reviewer for pointing out that our use of this term was not clear. We have revised the manuscript to clearly define this term (page 5, lines 6-16; page 7, line 5), and we have used this term consistently throughout the paper.

3.5 5, 13 What is meant by “susceptibility”? Here “intrinsic resistance” is defined partly, but after the term was already introduced above. Fig S4 should be in main paper.

See response 3.4 to clarify our use of the term ‘intrinsic resistance’. To avoid confusion, we replaced “antibiotic susceptibility” with the term “intrinsic resistance” (page 7, line 5).

This was a major experimental study, and we produced more data than can be accommodated in a typical paper in a high profile journal like Nature Communications. The key data presented in Figure S4 (resistance measurements, as IC50) are also presented in Figure 1C (correlation of IC50 and evolvability).

3.6 Line 23 seems to contradict lines 17, 18.

After carefully re-reading these sentences, it is not clear to us what the contradiction is.

3.7 7, 5 Assume “from each of 121 parental strains” is meant.

The Reviewer #3 is correct. We changed the sentence as suggested above.

3.8 7, 14-20 The explanation for non-selection of not-target gene SNPs is good but clarify the

claim regarding silent versus replacement mutations. What do all the numbers represent? What numbers would be expected otherwise, if the non-target alleles had selective value?

In response to this helpful criticism we have clarified this section of the manuscript, and we have added a reference to a paper that gives an excellent overview of methods for testing selection in microbial genomes (reference 31, page 8, lines 3-10). Our test for positive selection compares the number of replacement to silent mutations in protein coding regions. Although the actual number of observed replacement mutations (n=33) is much higher than the number of silent mutations (n=3), the excess of replacement mutations (Ka/Ks) is not significantly higher than we would expect due to chance alone. This is because the codon usage of the *S. aureus* genome is such that random mutations have a very high probability of causing an amino acid substitution.

3.9 8, 4 Mutation rate or mutation frequency? Mutation rate is notoriously difficult to measure.

The reviewer is correct that mutations rates are much more difficult to measure than mutant frequencies (which are commonly reported in the *S.aureus* antibiotic resistance literature). We measured mutation rates using Luria-Delbrück fluctuation tests, as outlined in the methods (page 25, lines 3-17) and Supplementary methods.

3.10 8, 13 Now we learn that norA is a major part of intrinsic resistance. This information needs to come earlier, at the definition of intrinsic resistance, not postponed like a mystery novel.

We have carefully considered this criticism, but we have decided not to introduce the role of *norA* earlier in the manuscript. The revised title of the paper now highlights the role of antibiotic efflux in the evolution of resistance, and the abstract emphasizes the important role of *norA*. Please also see our comment 2.4 explaining why we prefer to present experimental evolution and transcriptome study in a chronological order.

3.11 9, 5-10 There is much previous evidence that resistance gene amplification (Andersson; Nicoloff heteroresistance) lead to drug-resistant pathogens.

We agree with Reviewer #3 that there is substantial work has been done on the role of gene amplification in antibiotic resistance context. In fact, we cited two papers by D. Anderson (ref. 36, 37).

3.12 It is likely that highly variable copy number would not quantitatively parallel the degrees of drug resistance.

It is not clear to us why a disconnect between gene amplification and resistance should be our a priori hypothesis. However, our data support the assertion made by the reviewer (page 9, line 20-22)

3.13 As for “detectable costs of gene amplification,” there is evidence in other systems of high costs of gene amplification, both due to the translation costs and the costs of function of the expressed product.

We agree with Reviewer #3 that gene amplification has previously been shown to carry fitness costs, and we cite evidence of this in our manuscript (page 10, line 2-6, and reference 37).

3.14 The authors need to explain how they set up their fitness assay, not just show a figure.

We thank the reviewer for raising this point, and we apologize for this omission. We have now introduced a new section in the Supplementary methods to describe how this data was collected and analysed (Supplementary methods, Section 24).

3.15 Note that MIC is a poor indicator of relative fitness. Coculture competitions are required to demonstrate relative fitness differences at levels of drug 1/100 that of MIC.

In general, we agree with the reviewer that co-culture experiments provide a better way to assess relative fitness than mono-culture growth experiments. In the methods section of revised manuscript (page 27, lines 17 until line 4 on page 28) and in Supplementary methods (Sections 22 and 24), we provide more detailed information on the methods used to measure mono-culture growth rates, which we used as a proxy for fitness. In Figure 5B there is a very clear difference in growth rate under ciprofloxacin treatment (1mg/l) between *grlA* mutants under basal and elevated levels of *norA* expression. In this case, it is clear that co-culture studies are not required to measure the benefit of *norA* expression. In Supplementary figure 8, we show that evolved CC398 with *norA* amplification do not tend to have lower growth rates than their respective parental strains in the absence of ciprofloxacin. Although it is true that the sensitivity of this assay could have been increased by performing co-culture experiments instead of growth rate assays, it is very likely that our growth rate assays would have detected large costs of gene amplification, as have been previously reported in the literature (see point 3.11). Given that the results presented in Supplementary figure 8 are supplementary findings that are not required to establish the key findings of this study, we have decided not to undertake laborious co-culture competition experiments.

3.16 17, 8-10 Microplate evolution protocols are subject to cross-transfer between wells. The authors should address how they avoided crossover, and whether any evidence of crossover was seen in their results.

The Reviewer #3 is correct that cross-contamination represents a problem for transfer experiments using microtiter plates. To control for potential contaminations, 36 wells out of 96 wells per plate were not inoculated with bacteria. During experimental evolution only 2/6406 control measurements had optical density indicating contamination (this number in the denominator includes OD measurements for all transfers) (see Section 4 of Supplementary methods). However, during genomic sequencing, some samples did not match their expected parental genotype. These mismatches could have arisen due to either contamination during the experiment or due to the fact that some of the parental isolates were, in fact, polyclonal (we have found some evidence of this). In any case, discordant samples including other replicates from the same parental strain were completely excluded from the analysis.

REVIEWERS' COMMENTS:

Reviewer #1 (Remarks to the Author):

The authors have addressed the comments raised to my satisfaction.

Reviewer #3 (Remarks to the Author):

The authors report how a multi-drug efflux pump potentiates evolution of target-based drug resistance. The most interesting aspect of the work is the application of experimental evolution to clinical strains.

The NorA interpretation however is problematic.

Note: My comments below were written before I looked at the previous reviews. I now see that Reviewer 3 raises many of the points I did, which remain to be addressed.

Presumably the pump effluxes the drug, lowering the intracellular concentration, and thus allowing sufficient growth for the bacterium to acquire resistance in the drug target gene. The authors cite various published examples. Why are the authors trying to argue against this mechanism?

5, 1-3 Clarify "fraction of populations that survived." During evolution? MIC assay? How measured? This seems a problematic measure. Was it reproducible? Statistics?

5, 3-5 Clarify these sentences. Is the meaning that in 24 specific ancestral strains, resistance always evolves; whereas in 39 other ancestors resistance never evolves?

5, 16. Clarify what is shown in Supp. Fig. 4. What is MLST? Error bars or ANOVA?

5, 17 Clarify that "evolvability" means "evolvability of ciprofloxacin resistance." It seems trivial that strains already showing ciprofloxacin resistance will of course more readily evolve further resistance, as they are growing better.

6, 3-4. Explain how it was shown that "there was no correlation between the resistance of evolved populations and their parental strains." Also, why are all these figures supplementary, if they have primary importance?

7, 4. Were any of the strains hypermutators? This point should be addressed explicitly.

9, 3-6 The *norA* amplification is interesting. Does the amplification tend to be lost when ciprofloxacin is removed? 10, 1-6 Please explain how Supp Fig. 8 shows that amplification has no fitness cost.

12, 5 "norA overexpression was not actually sufficient to increase the MIC" This would be a good place to cite Gullberg and Andersson's work that antibiotics affect fitness at concentrations much lower than the MIC. Thus one should look for the MSC, minimum selective concentration.

14, 1-8 This argument is not convincing. Please address Andersson's work on the MSC. Tiny selective effects can indeed drive evolution.

15, 10-11 Why is this point inconsistent with *norA* acting by effluxing ciprofloxacin?

20, 12 Yes, Nolivos shows that AcrAB pump enhances acquisition of Tet resistance—by lowering Tet concentration so the plasmid-borne resistance genes can get expressed.

21, 1-13 The authors might say more about their microplate techniques and how they avoided cross transfer of evolving populations. Did they verify that all isolates appear independent, or did some cross transfer occur? This is a common issue with microplate evolution, so it would be helpful to address for the benefit of others attempting replication.

23, 10-11 Please explain the custom pipeline. Is the authors' code available? Pipelines make a big difference for what is found in resequencing. The authors should provide their code.

Reviewer #1 (Remarks to the Author):

The authors have addressed the comments raised to my satisfaction.

Reviewer #3 (Remarks to the Author)

:

The authors report how a multi-drug efflux pump potentiates evolution of target-based drug resistance. The most interesting aspect of the work is the application of experimental evolution to clinical strains.

The NorA interpretation however is problematic.

Note: My comments below were written before I looked at the previous reviews. I now see that Reviewer 3 raises many of the points I did, which remain to be addressed.

Presumably the pump effluxes the drug, lowering the intracellular concentration, and thus allowing sufficient growth for the bacterium to acquire resistance in the drug target gene. The authors cite various published examples. Why are the authors trying to argue against this mechanism?

We argue against this mechanism because our data show that efflux pump activity alone does not provide increased survival under antibiotic treatment. This point is addressed clearly in the manuscript (page 13, last paragraph; figure 5a).

5, 1-3 Clarify “fraction of populations that survived.” During evolution? MIC assay? How measured? This seems a problematic measure. Was it reproducible? Statistics?

Population survival was defined as an OD value of greater than 0.08, as defined in the methods. The main text (page 5, paragraph 1) clearly explains that this refers to the fraction of populations that survived until the end of the evolution experiment. The evolution of antibiotic resistance depends on selection for mutations that occur at random; thus, we do not expect evolution to be completely reproducible. However, our results show that some strains tend to have high evolvability, whereas other have low evolvability. For example, resistance always evolved in 24 strains(ie in 12/12 replicate populations), and resistance never evolved in 39 strains(ie in 0/12 replicate populations).

5, 3-5 Clarify these sentences. Is the meaning that in 24 specific ancestral strains, resistance always evolves; whereas in 39 other ancestors resistance never evolves?

Please see above.

5, 16. Clarify what is shown in Supp. Fig. 4. What is MLST? Error bars or ANOVA?

Supplementary Figure 4 shows different measures of the ciprofloxacin susceptibility (MIC, AUC, IC₅₀) of the 222 ancestral strains used in our experiment. The strains are colour coded according to their MLST, which is an important classification system used in bacterial population genetics based on sequence variation in conserved housekeeping genes.

Regarding error bars. In our methods, MIC is defined as minimal concentration in which no growth was observed in 3/5 replicates. As such, this metric provide only one value for all replicates, so errors bars are not applicable. Similarly, AUC was calculated simply as the area under dose-response curve produced by dose-response modeling. We have only one model fit per each strain and, therefore, only one value of AUC. In contrast, IC₅₀ was estimated as a parameter in dose-response analysis and it is possible to obtain standard error for this parameter.

In response to the reviewer concern, we now updated Supplementary Figure 2 and 4 to show standard errors for IC₅₀ and updated the legends to clarify how these measures were obtained.

5, 17 Clarify that “evolvability” means “evolvability of ciprofloxacin resistance.” It seems trivial that strains already showing ciprofloxacin resistance will of course more readily evolve further resistance, as they are growing better.

We have clarified that our measure of “evolvability” refers specifically to “evolvability of ciprofloxacin resistance” in the revised manuscript (page 5, paragraph 1). We agree with the reviewer that, all else being equal, strains with higher levels of intrinsic resistance should have greater evolvability (page 5, paragraph 2).

6, 3-4. Explain how it was shown that “there was no correlation between the resistance of evolved populations and their parental strains.” Also, why are all these figures supplementary, if they have primary importance?

We have clarified that our test compared the IC₅₀ of evolved populations and their ancestors (page 6, paragraph 1). This analysis suggests that the mechanisms of high level resistance that evolved during the selection experiment were independent of the intrinsic resistance of the parental strains. This assertion is supported by Figure 2A, where we

show that almost all strains evolved resistance mutations in a common set of well-defined ciprofloxacin targets (ie *grlA*, *griB*).

7, 4. Were any of the strains hypermutators? This point should be addressed explicitly.

We have added a point to our manuscript clarifying that we found no association between high evolvability and hyper-mutator strains (page 8, last paragraph).

9, 3-6 The *norA* amplification is interesting. Does the amplification tend to be lost when ciprofloxacin is removed?

We have not assessed the long-term stability of this amplification. However, given that amplifications are not associated with a detectable fitness cost, we do not expect rapid loss of amplifications.

10, 1-6 Please explain how Supp Fig. 8 shows that amplification has no fitness cost.

Supplementary figure 8 compares the growth rate of ancestral CC398 strains with their evolved derivatives carrying *norA* amplifications. Growth rate was measured in the absence of ciprofloxacin. If amplifications carry a cost, the evolved strains should have a lower growth rate than the ancestral strains. However, we found no tendency for evolved strains to have lower growth rates than the ancestral strains.

12, 5 “*norA* overexpression was not actually sufficient to increase the MIC” This would be a good place to cite Gullberg and Andersson’s work that antibiotics affect fitness at concentrations much lower than the MIC. Thus one should look for the MSC, minimum selective concentration.

14, 1-8 This argument is not convincing. Please address Andersson’s work on the MSC. Tiny selective effects can indeed drive evolution.

Both of these comments refer to an influential paper by that examines the impact of sub-MIC doses of antibiotic on selection for resistance strains (PMID: 21811410). The key insight from this paper is that low concentrations of antibiotic (ie well below the MIC of sensitive strains) are sufficient to generate selection for resistant strains). Our experiments address a fundamentally different scenario, where resistance evolves by *de novo* mutations under antibiotic concentrations that are above the MIC of the sensitive ancestral strain. As such, the link between the work of Andersson and Gullberg and our paper is unclear, and we have not added this citation.

15, 10-11 Why is this point inconsistent with *norA* acting by effluxing ciprofloxacin?

We agree with the referee on this point, and we have changed the text to refer specifically to the role of *norA* in ciprofloxacin efflux (page 15, paragraph 1).

20, 12 Yes, Nolivos shows that AcrAB pump enhances acquisition of Tet resistance—by lowering Tet concentration so the plasmid-borne resistance genes can get expressed.

This point is unclear, and this paragraph cites the recent paper by Nolivos *et al* examining the role of efflux pumps in plasmid acquisition (reference 74).

21, 1-13 The authors might say more about their microplate techniques and how they avoided cross transfer of evolving populations. Did they verify that all isolates appear independent, or did some cross transfer occur? This is a common issue with microplate evolution, so it would be helpful to address for the benefit of others attempting replication.

The key precautions to avoid contamination were:

- **30 out of 96 wells in each plate did not have bacteria to control for contamination. During experimental evolution only 2/6406=0.0003 control measurement showed $OD_{595} > 0.08$ indicating bacterial growth.**
- **Randomized design minimizes potential bias due to cross-contamination. For each strain, only 1 replicate culture out of 12 was located in the same microplate. If one microplate had accidental cross-contamination, it would affect only one replicate for a subset of strains located on that plate. (The experimental design involved more than 50 plates).**
- **Short duration of the experiment. We performed only 5 transfers with ciprofloxacin. Long-term experiments are more prone to contaminations.**

The detailed description of these methods is available Method section.

However, during genomic sequencing, some samples did not match their expected parental genotype. These mismatches could have arisen due to either contamination during the experiment or due to the fact that some of the parental isolates were, in fact, polyclonal (we have found some evidence of this). In any case, discordant samples including other replicates from the same parental strain were completely excluded from the analysis

23, 10-11 Please explain the custom pipeline. Is the authors' code available? Pipelines make a big difference for what is found in resequencing. The authors should provide their code.

We uploaded the code used for analysis to a public repository. This is now reflected in Code availability section.